# Caffeine induces age-dependent increases in brain complexity and criticality during sleep
Philipp Thölke [1,2] ✉, Maxine Arcand-Lavigne[1,2], Tarek Lajnef[1,2], Sonia Frenette[2,3], Julie Carrier [2,3] & Karim Jerbi[1,2,4,5]

Caffeine is the most widely consumed psychoactive stimulant worldwide. Yet important gaps persist in understanding its effects on the brain, especially during sleep. We analyzed sleep electroencephalography (EEG) in 40 subjects, contrasting 200 mg of caffeine against a placebo condition, utilizing inferential statistics and machine learning. We found that caffeine ingestion led to an increase in brain complexity, a widespread flattening of the power spectrum's 1/f-like slope, and a reduction in long-range temporal correlations. Being most prominent during non-rapid eye movement (NREM) sleep, these results suggest that caffeine shifts the brain towards a critical regime and more diverse neural dynamics. Interestingly, this was more pronounced in younger adults (20–27 years) compared to middle-aged participants (41–58 years) during rapid eye movement (REM) sleep, while no significant age effects were observed during NREM. Interpreting these data in the light of modeling and empirical work on EEG-derived measures of excitation-inhibition balance suggests that caffeine promotes a shift in brain dynamics towards increased neural excitation and closer proximity to a critical regime, particularly during NREM sleep.

For many, enjoying the delicate flavors of an espresso coffee ranks high up on the list of life's little pleasures. Unfortunately, although caffeine-containing products have a range of positive effects, such as increased alertness, mental focus, and cognitive performance, it also has a disruptive effect on the quality of sleep, which is critical for general health and well-being[1,2]. On the other hand, caffeine has been shown to exhibit neuroprotective qualities, particularly against Parkinson's disease[3–6], adding to the complexity of its impact on health. Caffeine is a psychoactive stimulant that is consumed by people across all age groups on a daily basis[7] through a wide variety of products such as coffee, tea, soft drinks, energy drinks, chocolate, and several pharmaceutical drugs[8]. It is therefore critical to understand how caffeine affects the brain during sleep, and across age.

Caffeine affects the quality of sleep in several ways. It increases sleep latency (i.e., the time it takes you to fall asleep) and it decreases sleep efficiency (the ratio of total sleep time to the time you spend in bed). The reduction of sleep duration caused by caffeine intake is particularly visible in the amount of time spent in the S2 sleep stage[9–12]. Further, both acute and regular daytime caffeine intake was found to delay rapid eye movement (REM) sleep promotion and lead to a reduction in the quality of

awakening[13]. However, a recent study proposed that the caffeine-induced reduction in sleep duration observed in humans could be attributed to a lack of flexibility in wake-up times instead of being a direct effect of caffeine[14]. Nevertheless, lack of sleep and sleep disorders can lead to the deterioration of the proper functioning of sleep-related brain processes, weight gain[15], hypertension[16], cardiovascular diseases, diabetes[17,18] and increase the risk of depression[19].

As a psychostimulant and an adenosine antagonist, caffeine reduces natural homeostatic sleep pressure by binding to adenosine receptors, thereby inducing a feeling of higher alertness and invigoration[20]. Adenosine, which is crucial for brain homeostasis, has been shown to mediate permeability of the blood-brain barrier (BBB) by binding to $A_1$ and $A_2$ adenosine receptors[21,22]. Activation of these receptors leads to increased permeability of the BBB, allowing transport of macromolecules like dextrans and $\beta$-amyloid antibodies into murine brains[21]. Hence, because it is an adenosine antagonist, caffeine is likely to interfere with BBB permeability rhythms during sleep. Despite encouraging progress, the mechanisms by which caffeine alters brain dynamics during sleep remain poorly understood. Beyond its direct effects on adenosine signaling, caffeine triggers a cascade of

[1]Computational and Cognitive Neuroscience Lab (CoCo Lab), Université de Montréal, Montréal, QC, Canada. [2]Psychology Department, Université de Montréal, Montréal, QC, Canada. [3]Centre for Advanced Research in Sleep Medicine, Research Center CIUSSS du Nord-de-l'Ile-de-Montréal, Montréal, QC, Canada. [4]MILA (Quebec Artificial Intelligence Institute), Montréal, QC, Canada. [5]UNIQUE Center (Quebec Neuro-AI Research Center), Montréal, QC, Canada. ✉e-mail: philipp.thoelke@posteo.de

downstream effects on other neurotransmitter systems, including enhanced dopamine and norepinephrine release, increased acetylcholine availability, and modulation of GABAergic and glutamatergic transmission[20]. These complex interactions between different neurotransmitter systems likely contribute to caffeine's diverse effects on brain function and sleep architecture.

## Electrophysiological effects of caffeine: from spectral power to brain complexity and criticality

Previous research has reported caffeine-induced effects on spectral power of electroencephalography (EEG) signals recorded during sleep. Generally, caffeine was found to decrease power in low frequency oscillations in the delta band and increase power in sigma and beta frequencies[9,10,12,23–26]. Another study conducted in Cynomolgus monkeys, which have been suggested to have sleep patterns similar to those of humans[27,28], found that caffeine ingestion led to a decrease in delta and theta power (1–8 Hz) and increased beta and low-gamma power (20–50 Hz) during sleep[29].

While spectral power analysis provides important insights into the oscillatory properties of brain signals, it does not capture the diversity and full complexity of EEG signals. Complementary insights can be harnessed by assessing brain signal complexity. The complexity of a system can generally be divided into two main subtypes: Type 1 complexity increases linearly with randomness (e.g., entropy or Lempel-Ziv complexity), while Type 2 complexity follows an inversely parabolic pattern, peaking at moderate randomness and declining at extremes (e.g., criticality or Kolmogorov Complexity)[30,31].

One common approach to measure Type 1 complexity is by measuring entropy, which generally assess the degree of unpredictability or randomness within a signal. These measures can provide valuable information about the underlying dynamical processes of the brain with high entropy signals occurring in states of wakefulness, while low entropy signals can be observed during deep sleep or anesthesia[32]. In fact, entropy was shown to vary consistently across sleep stages with wakefulness exhibiting the highest levels of entropy, followed by REM and NREM sleep[33]. Crucially, brain entropy has also been found to correlate positively with an array of cognitive functions including attention, memory, and verbal fluency[34].

However, Type 1 complexity as measured via entropy does not capture the full spectrum of complexity, as a purely random process–despite being highly entropic–lacks structure and meaningful organization. High entropy alone does not necessarily indicate maximal complexity in brain function. In contrast, Type 2 complexity addresses this limitation by highlighting that maximal complexity arises in systems that achieve an optimal balance between order and randomness. According to criticality theory, this critical point characterizes the state of maximal computational efficiency and optimal information processing. Such states, often associated with criticality, enable the brain to balance stability and flexibility, a hallmark of higher-order cognitive processes[35–37]. As a matter of fact, the so called edge of chaos criticality, characterized by a balance between order and disorder, can be assessed through various metrics, including Lempel-Ziv complexity, the slope of the aperiodic power spectrum, and long-range temporal correlations[35,38]. These measures have been shown to track cognitive states and arousal levels, suggesting they capture functionally relevant aspects of neural dynamics.

## Aim and hypotheses

The aim of the present study was to provide an in-depth assessment of the influence of caffeine on the brain's electrophysiological signals during sleep with a focus on brain complexity and criticality. Additionally, we expanded on previous power spectral investigations by disentangling periodic and aperiodic components in the EEG spectra, revealing a refined perspective on caffeine's impact on the EEG power spectrum during sleep.

In wakefulness, caffeine ingestion facilitates alertness and cognitive performance by blocking the action of adenosine, a neurotransmitter that promotes sleep drive. Given that cognitive performance is closely linked to brain complexity and criticality[37], one would expect that caffeine intake leads

to an increase in EEG complexity or entropy and a shift closer to a critical regime. In fact, a previous fMRI study has provided evidence for caffeine-induced increases in brain entropy during wakefulness[39]. However, it is yet unknown whether similar effects extend to sleep EEG signals.

Furthermore, caffeine is known to alter sleep architecture by reducing slow-wave sleep (SWS) and increasing lighter sleep stages such as N1 and N2[9,10]. Complexity measures, including entropy, have been shown to reliably reflect these changes, with entropy being highest in wakefulness, followed by REM sleep, and progressively decreasing across N1, N2, and SWS[32,33]. We expect that caffeine-induced alterations in sleep architecture, specifically the reduction of SWS and the increase in lighter sleep stages (N1 and N2), will result in a measurable increase in EEG complexity during NREM sleep.

Aging is characterized by a decrease in adenosine receptor density, reduced time spent in deep sleep, and shifts in brain dynamics, including increased neural entropy and a flattened power spectrum slope[33,40–42]. Given these baseline changes and caffeine's known effects on arousal and sleep architecture, we anticipate that caffeine's impact on brain complexity and criticality during sleep will be weaker in middle-aged compared to younger individuals.

Based on the above, the main hypothesis of this study is that caffeine ingestion leads to increased EEG complexity and a shift closer to a critical regime during NREM sleep. Our secondary hypothesis is that caffeine-induced changes in brain complexity and criticality are weaker in middle-aged compared to younger individuals, reflecting known age-related differences in adenosine receptor density, sleep architecture, and neural dynamics.

By contrasting the effects of caffeine and placebo using inferential statistics and machine learning (ML) separately for non-REM (NREM) and REM sleep, as well as for young and middle-aged groups of adults, we provide the first evidence that caffeine induces a broad increase in EEG brain entropy and a shift towards critical dynamics during sleep. These effects were more widespread in NREM compared to REM sleep, while age-related differences were observed exclusively in REM. To ensure the robustness of our results, we provide a comparison of several different metrics of entropy and criticality. Collectively, these findings advance our understanding of how caffeine modulates brain dynamics across different sleep stages and age groups, while offering methodological insights into the assessment of brain signal complexity and criticality under the effect of psychostimulants.

## Results

Sleep EEG data were collected from 40 healthy participants during two non-consecutive nights under caffeine and placebo conditions. After pre-processing and artifact removal, we extracted a range of features from the EEG, including power spectral density (PSD), entropy measures, and complexity metrics, to capture caffeine-induced changes in brain activity (see Section Feature extraction). These features were analyzed both statistically (see Section Direct statistical analysis) and using supervised ML classifiers (see Section Supervised machine learning analysis) to identify differences between conditions. A summary of Cohen's d values for statistical results can be found in Supplementary Table S3. The detailed experimental design, feature extraction, and analysis pipelines are described in the Methods section. In the following, we will go over the results of the spectral power analysis, followed by complexity and criticality-related observations, before covering age-related effects.

## Caffeine-induced changes in sleep EEG oscillations

Figure 1 shows the effect of caffeine, compared to placebo, on spectral power in five key frequency bands, compensating for changes in the aperiodic component (see Section Feature extraction). The topographies depict the results of standard statistical testing (permutation T-tests) alongside ML decoding accuracy topographies (SVM and LDA classifiers), and are presented separately for NREM and REM sleep stages.

During NREM sleep, caffeine led to statistically significant reductions in power in the delta, theta and alpha frequency bands (3 left columns in

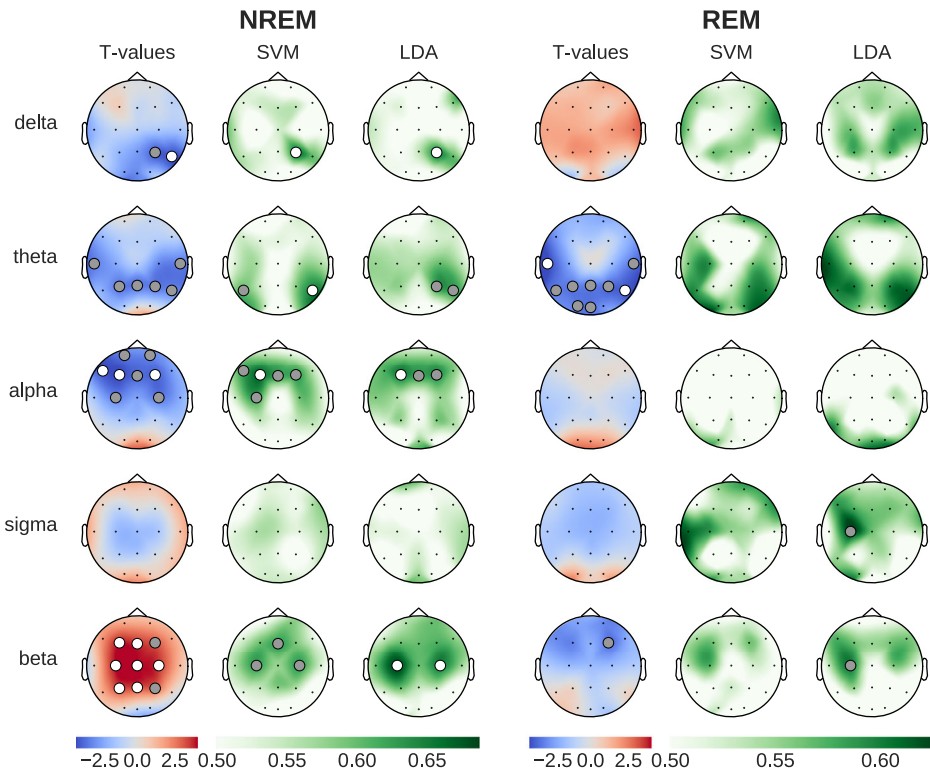

**Fig. 1 | Brain activity patterns during sleep (NREM and REM), comparing caffeine versus placebo effects on periodic neural oscillations (after removing aperiodic spectral components).** Left column shows statistical differences (blue: reduced during caffeine, red: increased during caffeine), while SVM/LDA columns show classification accuracy between conditions (green). Dots indicate statistical significance (gray: $p < 0.05$, white: $p < 0.01$).

Fig. 1). While this decrease spanned parietal and central channels for the delta and theta bands, the decrease in alpha power was localized between central and prefrontal channels. Interestingly, beta power showed a widespread increase across parietal and frontal channels. Compared to the results of the SVM and LDA classification, the t-test analyses yielded a larger number of channels with statistically significant results. This said, the spatial distributions of the significant results were consistent across all three methods, which speaks to the robustness of the findings across within-sample and out-of-sample analysis methods.

The most prominent caffeine effect observed during REM sleep was a significant reduction of theta power in temporal, parietal, and occipital channels. While the t-tests showed statistically significant differences between caffeine and placebo, the decoding accuracy obtained with SVM and LDA, although elevated, did not reach the $p < 0.01$ significance threshold level. This said, the LDA-based analysis during REM sleep revealed statistically significant ($p < 0.05$) classification using beta power in the frontal cortex. This effect was driven by a reduction in beta power, which contrasts with the strong beta power increase observed during NREM sleep.

**Disentangling contributions from periodic and aperiodic components to caffeine-induced spectral changes**
Crucially, and in contrast to previous work, the caffeine-induced power modulations discussed above were obtained using spectral feature computations that correct for the $1/f$ slope in the power spectrum (see Section Feature extraction). This correction takes into account the possibility that the aperiodic component of the spectrum ($1/f$ slope) may be different across the two experimental conditions. In which case, not correcting for the $1/f$ slope may lead to confounding results that amalgamate oscillatory and non-oscillatory signals. To appreciate the impact of this correction on the EEG presented here, we repeated the spectral power comparisons (caffeine vs placebo) shown in Fig. 1 but without correcting for the $1/f$ slope. The results of both approaches are shown in Fig. 2a. While the direction of the effect is largely consistent across both approaches (with the exception of delta power during REM sleep), we found the effects to be more pronounced after removing the aperiodic component from the power spectrum (channels with statistically significant $t$-values are scarce without the correction):

Several frequency bands do not show any significant effects when looking at the full original power spectrum and only become significant after removal of the aperiodic component.

Figure 2b depicts an illustrative example of power spectra from a single subject (dashed lines) and the fitted aperiodic components (solid lines), which are straight lines in a log–log plot. A flattening of the aperiodic component after the ingestion of caffeine is visible (see Section Caffeine-induced changes in sleep EEG complexity and criticality).

Taken together, the comparisons between the corrected and uncorrected spectra highlight the utility of removing the aperiodic component when assessing caffeine's impact on neural oscillations during sleep. In addition, the differences we observed between the results obtained with and without this correction strongly suggest that the aperiodic component of the spectrum (i.e., its slope) is affected by caffeine—otherwise removing it would not have modified the results. These changes in the scaling behavior of the EEG spectra can be linked to altered EEG self-similarity, which in turn may be related to shifts in neural criticality. In general, the discrepancies we found between the corrected and uncorrected spectral power results point towards the relevance of considering the aperiodic component of the power spectrum as a candidate EEG feature that is altered by caffeine.

**Caffeine-induced changes in sleep EEG complexity and criticality**
To assess the differential impact of caffeine and placebo on EEG complexity and criticality during sleep, we contrasted the two conditions using statistical inference and supervised ML, similar to our assessment of spectral power changes. The results shown in Fig. 3 depict the t-values and ML decoding accuracy obtained with multiple metrics in NREM and REM sleep stages. Overall, all EEG complexity measures (SpecEn, SampEn, SpecSampEn, and LZc) showed consistent increases under caffeine ($p < 0.01$ corrected, Cohen's $d$ between 0.67 and 0.93). In addition, the two metrics we used to probe shifts in critical behavior (i.e., DFA scaling exponent and the slope of the aperiodic activity) both showed consistent reductions, indicative of a shift towards the critical regime. During NREM sleep, the caffeine-induced modulations in brain signal complexity and criticality were widespread and prominent. However, during REM sleep, the effects were limited to the occipital regions and remarkably weaker (Fig. 3). More globally, the

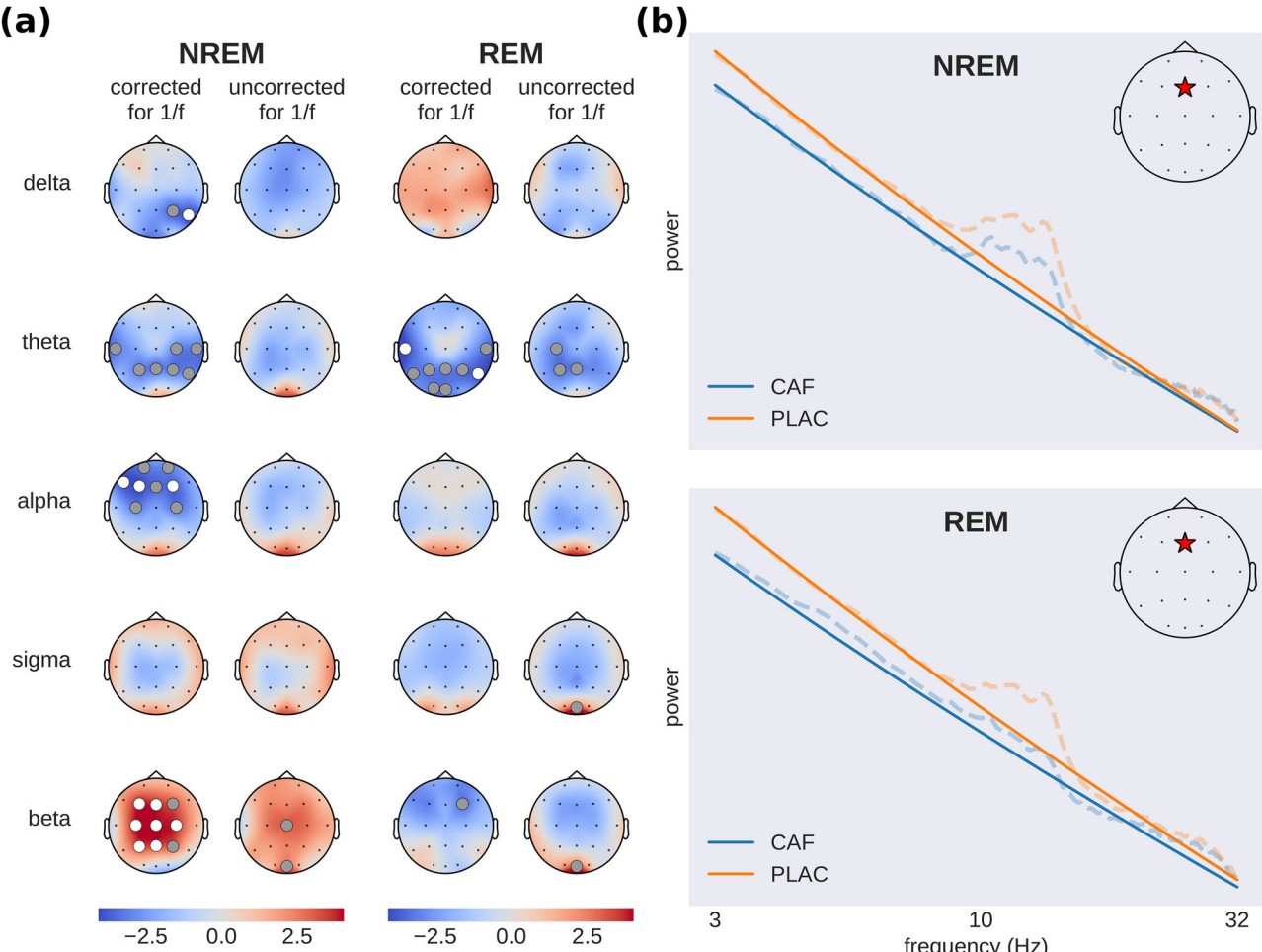

**Fig. 2 | Assessing the impact of changes in the slope of the aperiodic component on spectral power in canonical frequency bands. a** Brain activity before (uncorrected) and after (corrected) removing the aperiodic (1/$f$) component from the power spectrum. The topographic maps show t-values (caffeine-placebo condition) with blue indicating reduced and red increased spectral power during caffeine (dots: gray $p < 0.05$, white $p < 0.01$). **b** Illustrative example of caffeine-induced shifts in

aperiodic slope (solid lines), showing full power spectra (dashed lines) from a single subject at electrode Fz (red star) in a log–log plot. Subject and channel were chosen to be representative of the effect found across subjects. This panel does not contain results of statistical tests but rather serves as a clarification to the reader about the effect of the aperiodic slope on power spectral density across frequency bands.

direction of the effects was consistent among the four measures of complexity, and among the two metrics used to assess critical behavior. Interestingly, the most accurate classification between caffeine and placebo sleep EEG was obtained by using Spectral Sample Entropy as a feature in NREM sleep, with a 75% decoding accuracy, surpassing the best results from using spectral power. It is also worth mentioning that the mean DFA scaling exponent during NREM and REM sleep took values of 1.31 and 1.21, respectively.

**Single-epoch classification**
The features used in the classification analyses discussed so far were based on computing mean values at each electrode in each subject. We decided to extend this by asking whether it is possible to train a classifier to discriminate between caffeine and placebo conditions using features computed from single epochs, without averaging. The findings were largely inline with the results observed with the subject-wise averaged features (Supplementary Fig. S1). This said, the decoding performances were lower because the single-epoch features were obviously noisier than the averaged features. Given the large number of samples, decoding scores in the single-epoch classification were found to be statistically significant ($p < 0.01$ corrected) in a broad selection of channels even though passing the statistical decoding chance level only marginally. The overall agreement between the single-epoch and subject-wise ML results speaks to the robustness of the

observations as well as to the single-trial sensitivity of the selected features to the effect of caffeine.

**Multi-feature Machine learning**
In order to further explore the insights that can be harnessed through a data-driven approach, we built a random forest (RF) classifier which we trained on all extracted features simultaneously (11 features × 20 channels = 220 total; see Section Supervised machine learning analysis). The spectral features were corrected for the aperiodic component. We chose the RF classifier as it provides straightforward access to individual feature importance scores, allowing us to rank and compare the contributions of the distinct features in discriminating between the caffeine and placebo data. The RF classifier achieved substantially higher decoding accuracy in NREM sleep (75.22% ± 0.15) than in REM sleep (58.63% ± 0.13) after averaging the scores from 1000 random reinitializations of the classifier.

More importantly, the feature importance scores from the RF model shown in Fig. 4 revealed interesting insights into the nature of the most relevant features as well as their topographical distribution. Remarkably, the feature importance bar plots (colored according to the type of feature) reveal that, among all features, the complexity features are the most useful for the caffeine versus placebo classification during NREM sleep (all the blue features). These were followed by the criticality-related metrics (green), and then the corrected spectral power (warm colors). A closer look reveals that

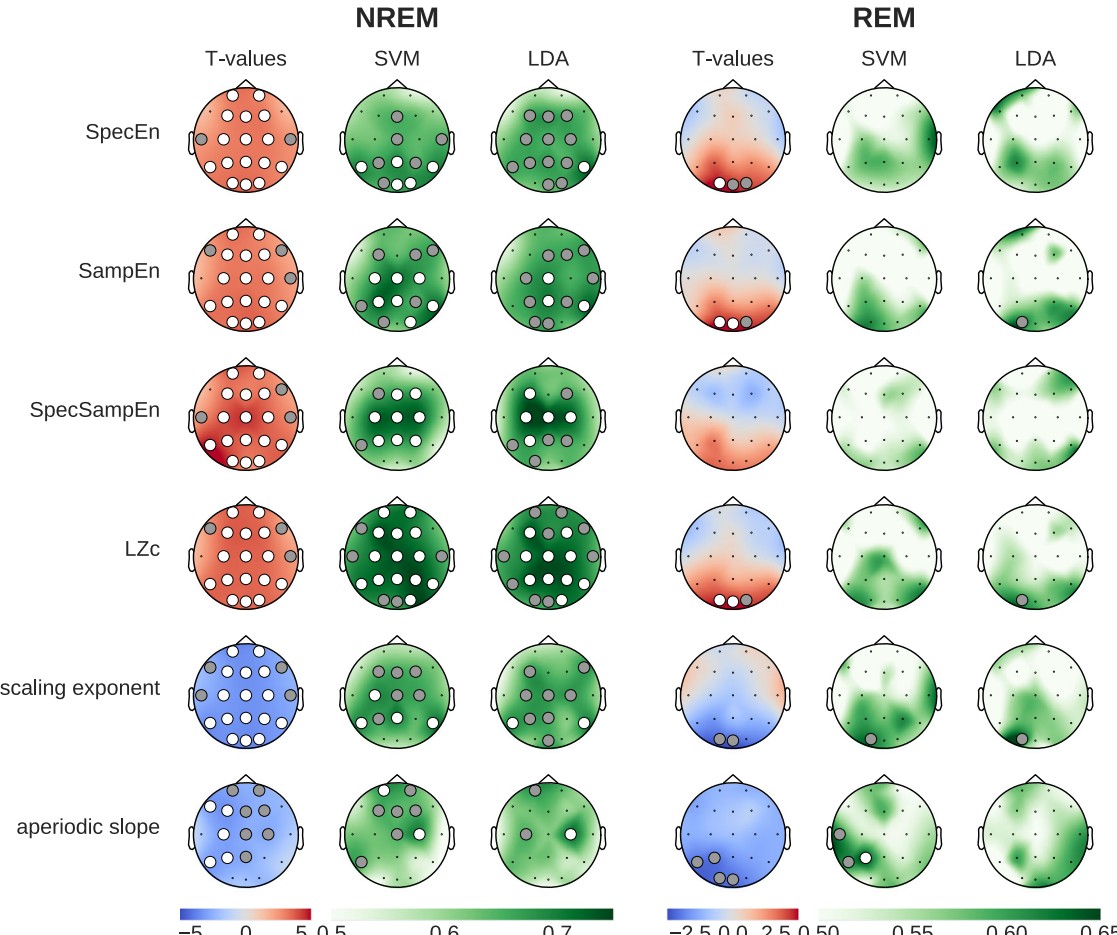

**Fig. 3 | Comparison of caffeine versus placebo effects on brain complexity and criticality measures during NREM and REM sleep.** Left columns show statistical differences (blue: reduced during caffeine, red: increased during caffeine). Middle and right columns show classification accuracy (green) from SVM and LDA models, validated using permutation tests. Dots indicate statistical significance (gray: $p < 0.05$, white $p < 0.01$). Most prominently, we see broad caffeine-induced increases in entropy and complexity, and a flattening of the aperiodic slope.

LZc and SpecSampEn exhibited high degrees of feature importance during NREM, contributing 37% of the classifier decision across all 11 features. Additionally, SampEn and the DFA scaling exponent saw prominent levels of feature importance. In other words, we found that the complexity and criticality-related measures outperformed the spectral features in terms of their contribution to the classification of caffeine versus placebo samples during NREM.

By contrast, the distribution of features ranked by importance during REM was more heterogeneous and less structured, including high levels of importance in temporal theta power, occipital DFA scaling exponent, and SpecEn. Sigma power and the aperiodic slope additionally showed increased contribution to the classification during REM.

Supplementary Table S2 provides a list of the ten most important feature-electrode pairs as determined by the RF classifier. These results highlight the prominence and leading contribution of complexity (as measured by LZc and SpecSampEn) over central, parietal, and frontal areas.

## Age-related effects

To explore how caffeine impacts sleep EEG across different age groups, we analyzed the data separately for young adults (20–27 years old, mean age 22.8 years; $n = 22$, 10 females) and middle-aged adults (41–58 years old, mean age 50.6 years; $n = 18$, 9 females). Figure 5 presents the results of paired T-tests comparing caffeine and placebo conditions within each group, as well as the independent T-test results for age-related differences (($\text{caffeine}_{young} - \text{placebo}_{young}$) vs. ($\text{caffeine}_{middle-aged} - \text{placebo}_{middle-aged}$)).

During NREM sleep, caffeine had significant effects on several EEG features in both young and middle-aged adults, although the effects were more pronounced in the young group. Notably, the independent T-test results revealed no significant age-related differences in the impact of caffeine during NREM sleep, suggesting that the weaker significance in the middle-aged group is likely attributable to the smaller sample size.

In REM sleep, caffeine-induced changes were significant only in the young group, affecting features such as spectral entropy (SpecEn), sample entropy (SampEn), Lempel-Ziv complexity (LZc), and the DFA scaling exponent. By contrast, no significant effects of caffeine were observed in the middle-aged group during REM sleep. Interestingly, the independent T-tests demonstrated significant age-related differences in these same features, indicating that the effect of caffeine on REM sleep is more pronounced in younger adults compared to middle-aged individuals.

To further contextualize these findings, we examined the baseline EEG differences between the young and middle-aged groups using data from placebo nights only (Supplementary Fig. S2). This analysis revealed that middle-aged individuals exhibited significantly higher complexity (SpecEn, SampEn, and LZc), a lower DFA scaling exponent, and a flatter aperiodic slope compared to the younger group, particularly during REM sleep. These age-related baseline differences align closely with the caffeine-induced effects observed in the young group, supporting the hypothesis that aging alters sleep EEG dynamics in a manner similar to caffeine, potentially diminishing its impact in the middle-aged group.

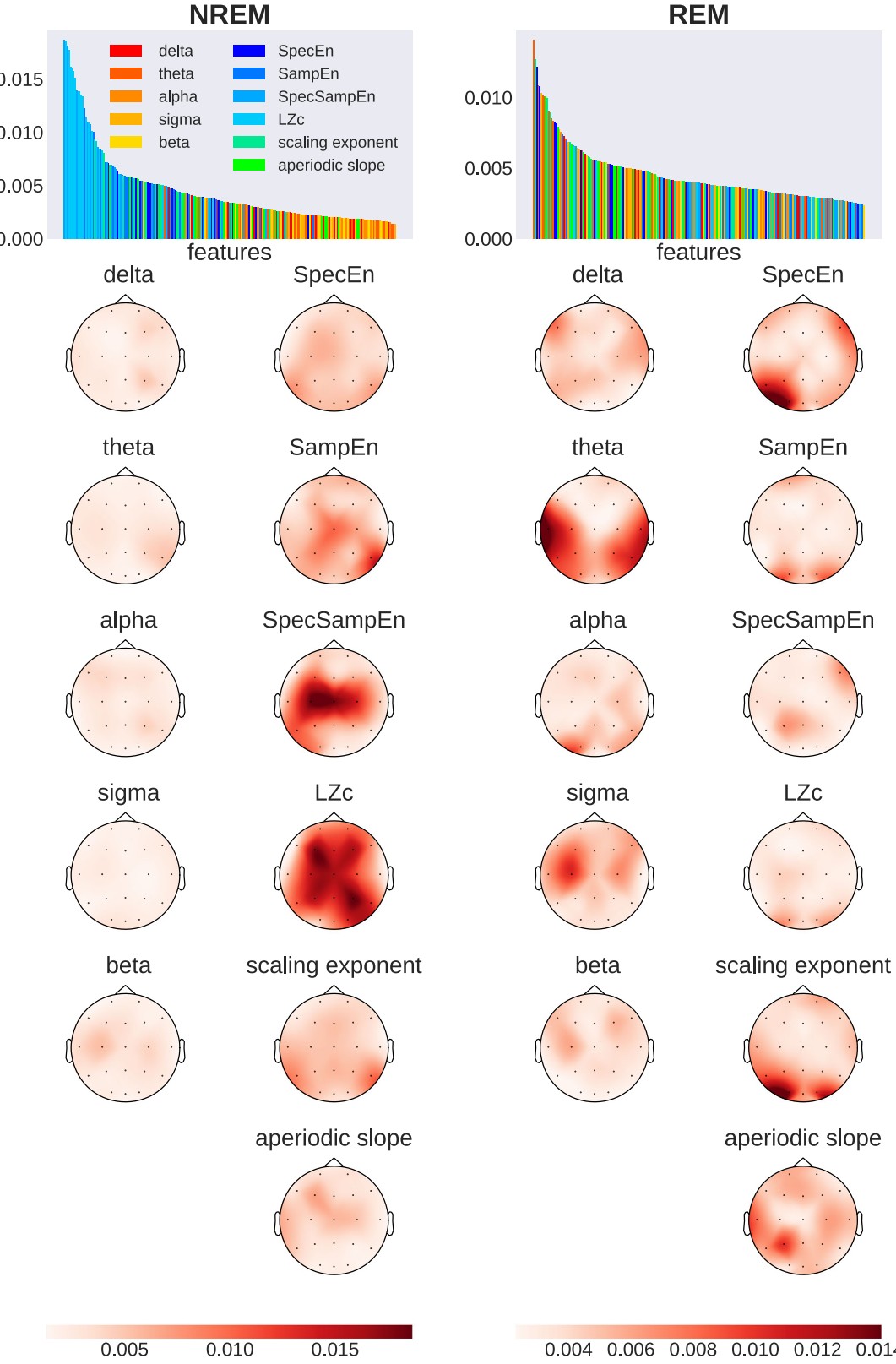

**Fig. 4 | Feature importance across brain regions during NREM and REM sleep, derived from random forest models trained on 220 features (11 features × 20 channels).** Bar plots rank input dimensions by feature importance, with warm colors (red to yellow) showing spectral power bands and cold colors (blue and green) showing measures related to entropy, complexity, and criticality. Topographic maps display the spatial distribution of importance values, averaged across 1000 models per sleep stage. Darker colors indicate higher feature importance.

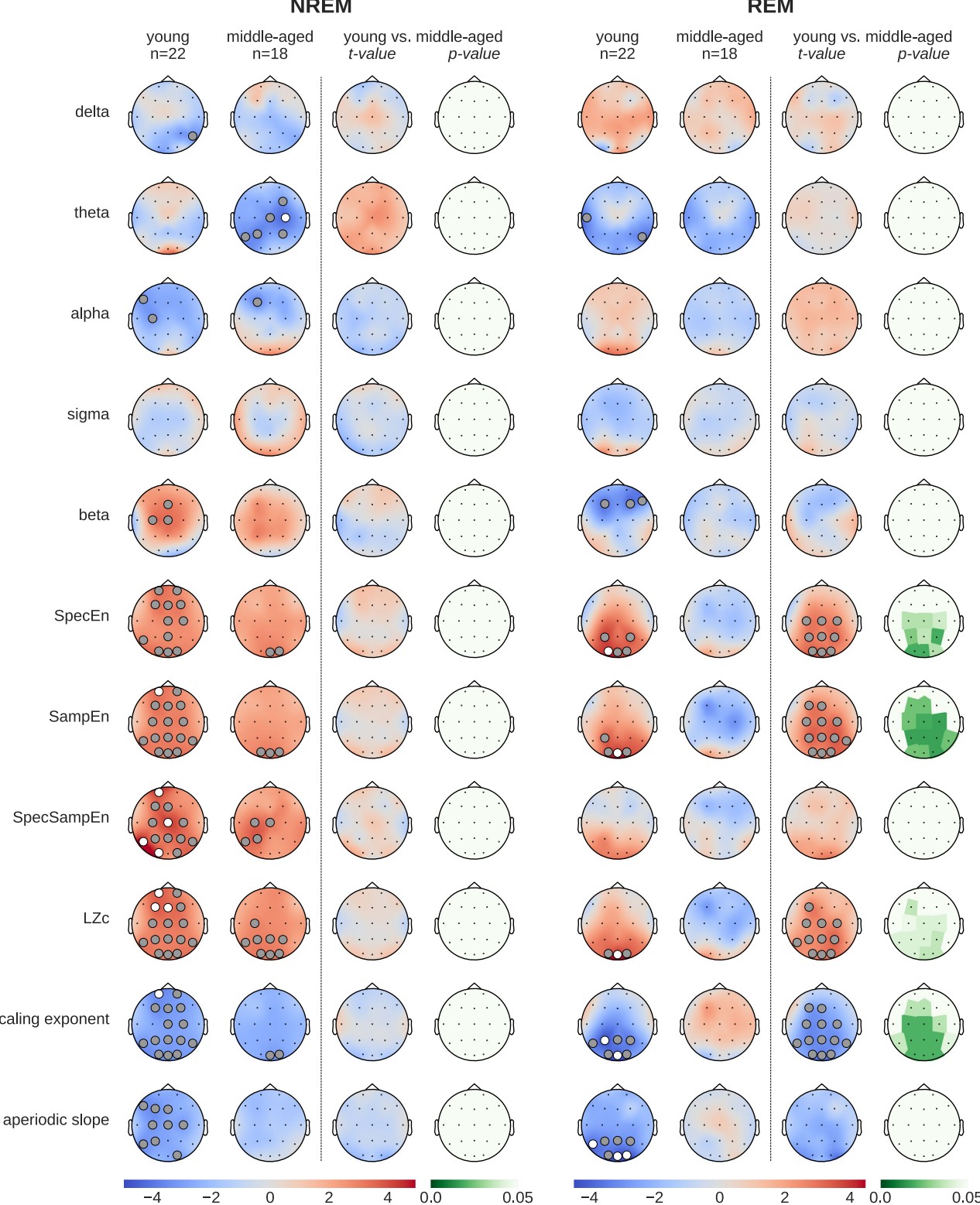

**Fig. 5 | Topographical maps showing age-related differences in brain responses to caffeine versus placebo.** The "young" (20–27 years) and "middle-aged" (41-58 years) columns display t-values of paired T-tests between caffeine vs. placebo. To rule out statistical effects of sample size, the remaining columns show the results of independent T-tests (t- and p-values) between age groups (($\text{caffeine}_{\text{young}} - \text{placebo}_{\text{young}}$) vs. ($\text{caffeine}_{\text{middle-aged}} - \text{placebo}_{\text{middle-aged}}$)). Gray dots indicate $p < 0.05$, white dots $p < 0.01$.

## Control analysis: sleep duration

Given that caffeine alters total sleep duration and architecture (distribution of sleep stages), it is theoretically conceivable that some of our findings were caused by an unequal number of epochs between caffeine and placebo.

Having more epochs in one condition would lead to more reliable averages due to reduced noise in the measurements. To rule out this possibility, we added a control analysis where we used random subsampling to equate the number of epochs used to compute the features across conditions. This was

done on a per-subject basis and separately for NREM and REM. This control analysis showed the robustness of the reported results and ruled out that they can entirely be attributed to caffeine-induced changes in sleep duration; we replicated the same statistical effects from Figs. 1 and 3 with an equated number of epochs in Supplementary Fig. S3.

## Discussion

The objective of this study was to investigate the effect of caffeine ingestion on the brain during sleep. In particular, we compared the effects of caffeine to that of placebo on neural oscillations, brain complexity and measures of critical behavior. Importantly, we set out to characterize these effects across REM and NREM sleep, and across age. In a nutshell, we found that caffeine induces a broad boost in brain complexity and a prominent shift towards criticality. These effects were substantially more widespread in NREM, compared to REM sleep. Furthermore, REM sleep in younger subjects (20–27 years) showed a significantly increased response to caffeine in terms of brain entropy when compared to middle-aged subjects (41–58 years).

Importantly, throughout this study, we paid particular attention to probing the robustness of our results through rigorous methodological procedures: First, the caffeine and placebo were contrasted using standard inferential statistics (t-tests) as well as two cross-validated ML classification algorithms (SVM and LDA) to check for consistency. Second, the statistical significance of these analyses (including the ML decoding results) was systematically assessed using non-parametric permutation tests and corrected for multiple comparisons. Third, we separated the aperiodic (1/f-like background) and periodic (oscillatory) components of the EEG power spectra before analysis, allowing us to specifically examine neural oscillations without the influence of broadband activity. Fourth, we used multiple metrics to characterize the underlying complex dynamics from different angles—examining entropy (signal predictability), complexity (information content and compressibility), and signatures of critical behavior—with largely consistent patterns emerging across these complementary measures. Fifth, we tested the robustness and interpretability of our ML framework by adding single-trial decoding as well as multi-feature classification using random forests, and feature importance assessment and visualization. Last but not least, we accounted for an array of possible confounds including age-dependent baseline EEG differences.

In the following, we discuss the key findings from this study in the light of previous work.

### An update on the impact of caffeine on neural rhythms during sleep

Our findings on power spectral density, when analyzed across the full power spectrum, align well with previous observations of caffeine's influence on EEG power in distinct frequency ranges. The existing literature highlights a decrease in delta and frontal alpha power and an increase in sigma and beta power, as well as in occipital alpha power[9,10,43,44]. Interestingly, when we used 1/f-corrected power spectra (i.e., excluding the aperiodic component), the results were more enhanced and reached statistical significance over numerous channels. As shown in Fig. 2a the caffeine-induced decrease in delta, theta, and frontal alpha power during NREM sleep only became statistically significant after removing the aperiodic component from the power spectrum.

In addition, discarding the aperiodic component uncovered a strong statistically significant increase in beta power over multiple parietal, central and frontal sites. While adenosinergic inhibition mostly targets excitatory neurons (specifically acetylcholine and glutamate), it also reduces GABAergic activity[45], a key modulator of beta oscillations in the brain[46–48]. Taken together, the caffeine-induced increase in beta power might be directly attributable to elevated GABA levels, which result from reduced adenosinergic inhibition of GABAergic neurons.

Moreover, while we saw a non-significant decrease in delta power during REM, matching previous reports[23], this effect appeared to be reversed when we used the 1/f-corrected power spectrum. This discrepancy—as well as the other differences we found using corrected versus uncorrected power

spectra—can be attributed to the 1/f activity acting as a confounding factor. Crucially, although commonly referred to as 1/f-noise, the aperiodic activity, has been shown to exhibit inter-subject variability that is linked to cognitive ability and age[41]. In sum, the caffeine-induced changes in oscillatory power reported here confirm but also extend previous reports by disentangling rhythmic components from broadband aperiodic activity.

### Caffeine ingestion increases EEG entropy during sleep

Prior quantitative work investigating the effect of caffeine on sleep EEG has predominantly focused on characterizing its effect on brain rhythms. Our findings extend this body of work by uncovering important effects of caffeine on the predictability of neural signals as measured via metrics of entropy. In fact, by analyzing the feature importance scores from random forest classifiers, we found robust evidence that EEG signal entropy played a more important role in distinguishing caffeine from placebo samples, compared to (corrected) spectral power features.

Although our results provide the first evidence that caffeine enhances EEG entropy during sleep, they are consistent with a previous fMRI study showing that caffeine enhances entropy during resting wakefulness[39]. Although widespread, the most significant entropy increases were found in the lateral prefrontal cortex, default mode network, visual cortex, and the motor network[39]. Although these fMRI results obtained in the waking state are not directly comparable to our polysomnographic sleep data, they are consistent with the caffeine-related increases we found across all entropy measures (SpecEn, SampEn, SpecSampEn) during NREM and REM sleep. Our findings extend the literature not only by revealing that caffeine elevates the complexity of the brain's electrophysiological activity, but also by detecting such changes during sleep. Finally, our data also suggest that this caffeine-mediated boost in brain entropy is age-dependent in the REM stage.

In a prior investigation assessing the complexity of biological systems, low approximate entropy (ApEn) was associated with the isolation and segregation of dynamical processes[49]. Furthermore, experimental work suggests increased segregation of the brain during NREM sleep[50,51], which was later shown to be reflected in decreased ApEn[52]. These observations provide an interesting framework in which to interpret the caffeine-induced increase in sample entropy (SampEn), which we found during NREM sleep. Given that SampEn is closely related to ApEn, our results may therefore indicate that caffeine ingestion reduces the separation between brain networks, promoting greater integration. These results also suggest an increase in information processing during NREM, which brain entropy is commonly seen as a proxy for ref. 53. This relationship between entropy and information processing aligns with theories of criticality in neural systems, where states near the critical point are characterized by maximal integration across scales and optimal information processing capacity[37]. The higher SampEn values we observed after caffeine administration may indicate a shift away from the subcritical regime, where local processing dominates and in which the brain operates during NREM sleep[54]. This shift towards the critical point may reflect a transition into a more integrated state that allows for enhanced information flow across multiple temporal scales, which also aligns with our findings on LRTC, as evaluated by the DFA scaling exponent.

Higher entropy in neural signals during sleep has been associated with enhanced information integration and dynamic adaptability, processes that are critical for cognitive functions reliant on sleep, such as optimal neural communication and adaptive information processing. The caffeine-induced increase in EEG entropy during sleep may therefore reflect changes to these processes, potentially impacting the brain's ability to efficiently process and integrate information across different neural states. As the caffeine-induced increase of brain entropy was most pronounced in NREM, the stage known to exhibit low brain entropy, it is tempting to associate the effect of caffeine with a deterioration of sleep quality. Although increased brain entropy during sleep has been linked to hypertension[55] and early-stage Alzheimer's disease[56], further investigation is needed to elucidate the impact of pharmacologically induced alterations in sleep brain entropy and their implications for sleep-dependent cognitive functions.

## Caffeine increases Lempel-Ziv complexity and shifts critical dynamics in the sleeping brain

Given that Lempel-Ziv complexity (LZc) quantifies the compressibility of neural signals—another measure of the richness of information content – it is not surprising to see that caffeine's influence on LZc and entropy metrics was similar, displaying widespread increases during NREM sleep and a localized increase in occipital channels during REM sleep compared to placebo. Interestingly, LZc has previously been shown to inversely track the distance to criticality in the brain, i.e., an increase in LZc is tied to activity closer to the critical point[35], which further supports the hypothesis that caffeine causes a shift from subcritical dynamics closer to the critical point, particularly during NREM sleep. LZc has also been found to be inversely related to the aperiodic slope, another marker for critical dynamics[57,58]. Additionally, the observed increase in entropy and decrease in DFA scaling exponent suggest elevated degrees of activity during NREM sleep[33,59].

Considering our results in light of the brain criticality literature suggests that—especially during NREM sleep—the dynamics of the caffeinated brain are shifted closer to the critical point and towards a more active state. Our data also suggest that this effect is more pronounced in younger subjects (20–27 years old) than in middle-aged subjects (41–58 years old), with age effects observed only during REM sleep, potentially due to aging-related changes in adenosine A1 receptor density. It is tempting to associate these findings with the deterioration in sleep quality and its restorative properties, as documented in the literature on caffeine and sleep.

## Caffeine-induced neural excitation is associated with a shift towards a critical point

Caffeine works by reducing the activity of adenosine, an inhibitory neurotransmitter that causes drowsiness. As a result, consuming caffeine enhances alertness, improves information processing, and boosts cognitive performance. However, due to adenosine's key role in regulating sleep, the effect of blocking adenosine receptors via pharmacological intervention might not be equivalent during wakefulness and sleep, as the underlying adenosinergic signaling and receptor dynamics likely differ between these distinct brain states. Furthermore, caffeine not only affects adenosine signaling directly but also triggers a cascade of changes in other neurotransmitter systems, including increased dopamine and norepinephrine release, enhanced acetylcholine availability, and modulation of the balance between GABAergic and glutamatergic transmission[20,45]. While these complex interactions across multiple neurotransmitter systems make it challenging to predict caffeine's global effects on brain dynamics, examining changes in the excitation–inhibition balance may provide a more tractable framework for understanding its impact.

Both modeling and experimental studies have shown that the slope of the $1/f$ component of the power spectrum (and related measures like scaling and Hurst exponents) can serve as indicators of the excitation-to-inhibition (E:I) ratio[57,58,60–63]. Our data reveal that caffeine induces a flattening of the $1/f$ slope and a drop in the scaling exponent, providing evidence for a shift towards increased excitation. Interestingly, this finding takes on additional significance within the framework of brain criticality, where E:I balance serves as a control parameter that can drive neural dynamics closer to or further from the critical point. The critical point represents a state poised between order and chaos that allows for maximal computational efficiency and flexibility[37]. Our results suggest that caffeine reduces the characteristic inhibition-dominated dynamics typically observed during sleep, particularly during NREM sleep, shifting the system towards a state of increased excitation.

Interestingly, the link we make here between caffeine's antagonistic effect on adenosine receptors and the observed shift in criticality is conceptually consistent with previous research showing that by blocking GABA receptors and thus reducing inhibitory synaptic transmission, an artificial upward shift in excitation can occur, leading to a supercritical state with larger-than-expected neuronal avalanches[64,65]. This suggests that, while adenosine blockers have an upregulating effect on GABAergic neurons, the global impact of caffeine on the excitation-inhibition (E:I) balance appears to be positive, as indicated in the literature[45].

## Age-related differences in the impact of caffeine on the sleeping brain

Previous research has shown that middle-aged adults are more sensitive to caffeine's effects on sleep latency, duration, and efficiency[10], but age-related differences in electrophysiological sleep features remain less understood. In our study, we observed that caffeine had a significantly greater impact on REM sleep EEG features (specifically SpecEn, SampEn and DFA scaling exponent) in younger participants compared to middle-aged adults, while no significant age effects were found during NREM sleep.

This pattern may reflect an interaction between age-related neurophysiological changes and the distinct ways in which adenosine modulates NREM and REM sleep. Adenosine is a key regulator of sleep-wake dynamics, and its effects are mediated primarily through $A_1$ and $A_2$a adenosine receptors[66]. Aging is associated with a natural decline in $A_1$ receptor density[67], which likely reduces the capacity of adenosine to modulate sleep-related processes, particularly during REM sleep, where adenosine activity is already lower compared to NREM sleep[66].

In younger adults, higher $A_1$ receptor density may amplify the effects of caffeine, which functions by antagonizing adenosine binding; this amplification may be attributable to the increased receptor availability, thereby promoting a higher rate of receptor inhibition. During REM sleep, this interaction could result in a compound effect of reduced adenosine activity (characteristic of REM) and higher receptor availability, enabling caffeine to exert a more pronounced influence on brain dynamics. By contrast, the diminished $A_1$ receptor density in middle-aged adults likely limits caffeine's impact, as fewer receptors are available for adenosine binding and subsequent blockade by caffeine.

The absence of significant age effects during NREM sleep may also be explained by the interplay of these factors. Adenosine activity is upregulated during NREM sleep[66], and this robust baseline activity may mask age-related differences in receptor availability. Consequently, while caffeine induces comparable changes in NREM-related EEG features across age groups, the distinct dynamics of adenosine and receptor density during REM sleep appear to drive the observed age-dependent effects.

Beyond receptor mechanisms, several factors may explain age-related differences in caffeine sensitivity during REM sleep. Age-related changes in caffeine metabolism[68] likely play a role, as decreased hepatic clearance in older adults can alter caffeine's concentration during different sleep phases, potentially explaining the attenuated effects we observed in middle-aged participants. Baseline sleep architecture differences may also contribute–middle-aged adults typically experience reduced REM sleep quantity and quality[69]—potentially creating a ceiling effect where already compromised REM sleep shows less disruption from caffeine. Additionally, age-specific lifestyle factors such as work stress, family responsibilities, and exercise habits may modulate baseline sleep characteristics and caffeine responses through indirect pathways involving stress hormones and inflammatory markers[70].

Taken together, these findings suggest that caffeine's greater impact on younger adults' REM sleep EEG features arises from an age-dependent interplay between adenosine signaling, receptor density, caffeine metabolism, and caffeine's pharmacological action. Age-related differences in hepatic clearance and baseline sleep architecture likely contribute to the attenuated response in middle-aged adults. Future studies should further explore these interactions, particularly with respect to regional and receptor subtype-specific variations in adenosine activity, caffeine pharmacokinetics, and lifestyle factors, to better understand the nuanced effects of caffeine on sleep across the lifespan.

## Limitations

A few limitations of the present study are worth considering. First, although the statistical comparisons and ML classification revealed significant effects in a substantially larger number of channels (and higher decoding

performances) in NREM compared to REM sleep, we need to keep in mind that the different number of epochs available for each stage in each subject can lead to comparatively noisier estimates of the extracted features in REM. This said, during preprocessing, we averaged features across epochs of each sleep stage in each subject, effectively reducing per-sample noise.

Second, despite the strong consistency between the effects we found across three entropy metrics—especially between SpecEn and SpecSampEn —we should be mindful that SpecSampEn is a slight variation of spectral entropy computation that is not commonly used. Therefore, its increased feature importance compared to SpecEn in the multi-feature random forest analysis (in NREM sleep) is a possibly interesting observation, but it will require further validation and needs to be interpreted with caution.

Third, while some of our interpretations rely on modeling and empirical evidence that the E:I ratio is correlated with the slope of the power spectrum and the DFA scaling exponent, more work is needed to fully establish the mechanistic links between these EEG measures and E:I balance at multiple meso- and macroscopic scales[46].

Fourth, caffeine is known to alter the ratio of NREM stages (N1, N2, SWS) to each other, making it difficult to disentangle the effect of caffeine on individual NREM stages from the caffeine-induced shift in sleep architecture (less SWS, more N1/N2). Many of the caffeine-related significant changes we found in NREM are characteristic of such a shift. Specifically, hallmark features of SWS, such as low entropy/complexity, will contribute less to the aggregated measures investigated in this study.

Finally, our study focused on healthy individuals, which may limit the generalizability of our findings. Altered baseline brain dynamics and sleep architecture associated with sleep disorders and neurodegenerative diseases may interact with caffeine's effects on the brain in complex ways. While regular daytime caffeine consumption has been associated with neuroprotective qualities, particularly in the context of Parkinson's disease[3–6], acute caffeine intake close to bedtime is known to disrupt sleep, potentially affecting critical brain processes that occur during sleep. For example, caffeine could hypothetically exacerbate sleep fragmentation already present in Alzheimer's[71] and Parkinson's disease[72], though our study does not provide empirical evidence for this specific interaction. Given the complex interplay between caffeine's beneficial neuroprotective effects, its disruptive influence on sleep, and the sleep physiology changes observed with aging, future research should examine how caffeine-induced alterations in sleep brain dynamics manifest across different clinical populations. Such studies could help inform tailored caffeine consumption recommendations for individuals with neurological disorders.

## Conclusion

In this work, we investigated the effects of caffeine, the most widely consumed psychoactive drug, on the activity of our brains as we sleep. In particular, we extend previous research on the neural impact of caffeine on sleep by focusing on two promising—yet underexplored—features of neural dynamics, which are complexity and criticality. This study provides the first evidence that caffeine ingestion leads to a broad increase in EEG complexity, especially during NREM sleep. Crucially, we also discovered that, compared to placebo, caffeine shifts the brain closer towards a state known as a critical regime, where the brain is thought to be most sensitive to inputs, most adaptable, and able to process information most efficiently. Importantly, we propose a mechanistic explanation of the observed shifts by connecting caffeine's effect on adenosine transmission to changes in E:I balance, which are manifest across multiple EEG measures. Finally, our data suggest that the shifts in brain dynamics due to caffeine are more prominent in younger adults than in middle-aged individuals during REM sleep, a difference that could be explained by aging-related changes in adenosine $A_1$ receptor concentration. While our primary focus was on the influence of sleep on brain electrophysiology, we anticipate that future studies will uncover similar outcomes during wakefulness. Considering the widespread consumption of caffeine, gaining a comprehensive understanding of its effects on the brain both during wakefulness and sleep could have far-reaching implications for society and public health.

## Methods and materials
### Data acquisition
Sleep EEG (256 Hz) data were recorded from 40 subjects (19 females, 21 males) at the Center for Advanced Research in Sleep Medicine (CARSM), with a channel layout according to the 10–20 international system. Subjects were in good health, aged from 20 to 58 (mean $35.3 \pm 14.3$ years). The participants reported moderate caffeine consumption, equivalent to one to three cups of coffee per day. All participants were non-smokers and free of drugs or medicine which could influence the sleep-wake cycle. Subjects also reported no sleep complaints, night work, or transmeridian travel in the 3 months before the recording. Participants with a history of psychiatric or neurological illnesses or a body mass index (BMI) above 29 were excluded from the study. Blood sample analysis (complete blood count, serum chemistry including hepatic and renal functions, levels of prolactine, levels of testosterone in men, and levels of estrogen, follicle-stimulating hormone (FSH), and luteinizing hormone in women) and urinalysis results were examined by a certified physician to exclude significant medical irregularities. To eliminate further irregularities in participants, a polysomnographic screening night at the laboratory was conducted where a nasal/oral thermistor, electromyographic leg electrodes, EEG, and electro-oculogram were recorded. The presence of sleep disturbances such as sleep apnoeas and hypopnoeas (index per hour >10), periodic leg movements (index per hour >10), prolonged sleep latency (>30 min), or low sleep efficiency (<85%) resulted in the exclusion of the participant. Peri-menopausal women and women using hormonal contraceptives or receiving hormonal replacement therapy were excluded, premenopausal women reported having regular menstrual cycles (25–32 days) during the year preceding the study, no vasomotor complaints (i.e., hot flashes, night sweats) and showed low FSH levels (<20 iU L-1) and all postmenopausal women reported an absence of menses during the past year and their FSH levels were >20 iU L-1. All participants signed an informed consent form which provided detailed information about the nature, purpose, and potential risks of the study. The research project was approved by the hospital's ethics committee, and subjects received financial compensation for their participation. All ethical regulations relevant to human research participants were followed.

### Experimental protocol
After the adaptation and screening night, each participant spent two non-consecutive experimental nights at the sleep laboratory. The time between each night in the laboratory was 6–9 days. On days of recordings, the consumption of foods and beverages containing caffeine was ceased at noon. Instructions were given to maintain a regular sleep-wake pattern within 30 min of the habitual sleep-wake cycle, as well as a habitual caffeine consumption one week before the first night to prevent potential withdrawal effects. They were asked to complete the French version of the Pittsburgh Sleep Diary[73] on a daily basis during this time and to abstain from alcohol on experiment days. The participants arrived at the laboratory 6–8 h before their habitual sleep time and left 1–1.5 h after habitual wake up time. Bedtime and wake time in the laboratory were determined by averaging each participant's sleep-wake cycle from the sleep diary. The total dose of caffeine administered was 200 mg (100 mg per capsule) which is considered to be moderate (equivalent to 1–2 cups of coffee) and induces significant changes in the sleep of young subjects[23]. Two-piece telescopic hard capsules were used, allowing the ingestion of caffeine without oral contamination, in a double-blind crossover design using stratified randomization.

### Sleep variables
An extensive analysis of sleep variables and their alteration with caffeine was conducted in previous work in our group with the same data[9,10]. Briefly, this data exhibits a significant caffeine-induced increase in sleep latency and a decrease in total sleep time, sleep efficiency, and time spent in stage 2 sleep[9,10]. We therefore controlled for the effect of changes in sleep duration in our analysis and found no significant change, suggesting our results are robust in this regard. In other words, the difference in the feature estimations

was not due to the variable amount of data across conditions but rather to the underlying changes in neural dynamics.

## EEG preprocessing

The data was divided into 20s windows and visually scored by an expert into five sleep stages: S1, S2, S3, S4, and REM according to the Rechtschaffen and Kales manual[74], modified to allow scoring based on 20s epochs[10]. While scoring sleep into S1, S2, S3, and S4 is no longer standard practice and has since been replaced by the AASM guidelines, which consolidate S3 and S4 into a single N3 stage, this approach reflects the common scoring practice at the time of data collection. Importantly, this distinction does not impact the study's conclusions, as we combined S1, S2, S3, and S4 into a single NREM stage for analysis. From a signal processing standpoint, we have no reason to believe that our results would differ were we to use 30s epochs. After scoring, artifact-containing epochs (4s) were eliminated from NREM data by visual inspection of channels C3 and C4. Due to the relatively low number of epochs in S1, S3, and S4 compared to S2, these stages were grouped into the broader NREM category. This is consistent with previous research conducted on the same data[9]. After artifact removal, the NREM stage contained 57,594 epochs, while REM consisted of 19,341 epochs, yielding a roughly 3:1 ratio. Before feature extraction, we applied a band-pass filter from 0.5 to 32 Hz to the raw epochs, removing further artifacts from the signal. Features were extracted independently for each channel, resulting in a single value per feature, channel, and epoch. To reduce noise and increase comparability between sleep stages, the epoch-wise features were averaged within each subject. For each feature, given that we had 40 individuals, this led to a total of 80 samples: 40 for caffeine and 40 for the placebo condition. Furthermore, the averaged features were z-transformed using the mean and standard deviation from the non-averaged samples across all channels and samples of one feature. The z-transform was applied separately to each subject, ensuring strict independence between training and test data sets in the ML analysis.

## Feature extraction

We extracted a selection of relevant hand-crafted EEG features from the neural signals to examine the effect of caffeine on the brain in a data-driven manner. The aim was to differentiate between data from the caffeine or placebo condition using supervised machine learning, trained on these features. For the sake of consistency, all features were extracted from 20s epochs of continuous EEG. The following sections list the specific spectral, complexity, and criticality features analyzed in this work. Note that while the critical point is formally defined only for infinite systems, finite systems can still operate in a critical regime, exhibiting approximate scale invariance and power-law relationships over a range of scales. Therefore, the criticality features discussed below, such as DFA scaling exponent and aperiodic slope, remain valid indicators of critical dynamics even in finite neural systems.

**Power spectral density**. Power spectral density (PSD) was computed using Welch's method[75] with a window length of 4s and an overlap of 2s. To isolate changes in periodic (pure oscillatory) activity, we remove the aperiodic $1/f$-like component from the power spectrum. Disentangling periodic and aperiodic components in the EEG spectra enables the isolation of distinct neural processes, with periodic activity reflecting oscillatory synchronization and aperiodic activity providing insights into scale-invariant properties of the EEG and the $1/f$-behavior of its power spectrum, which has been linked to the neural excitation and inhibition balance[57,58]. This distinction is particularly relevant for understanding how caffeine modulates brain dynamics, as changes in the aperiodic component may highlight shifts in cortical excitability, while periodic changes can reveal alterations in specific oscillatory rhythms critical for sleep and cognitive function. The FOOOF algorithm[76] was used to fit a power-law distribution to the power spectrum and extract periodic oscillations in the frequency range from 3 to 32 Hz, limiting the number of periodic peaks to 5. The lower and upper cutoffs were determined

visually from the full power spectrum. The $1/f$-corrected spectrum was then divided into five distinct frequency bands by averaging the power across frequency bins within the given intervals. The final frequency bands were delta (0.5–4.0 Hz), theta (4.0–8.0 Hz), alpha (8.0–12.0 Hz), sigma (12.0–16.0 Hz), and beta (16.0–32.0 Hz). We extrapolated the power-law distribution to also cover the delta band.

**Sample entropy**. To estimate complexity in the time domain, sample entropy (SampEn)[77] was computed using the preprocessed EEG signal. The definition of SampEn is similar to that of approximate entropy (ApEn), a popular metric for complexity estimation of biomedical times series. ApEn, however, is known to have limitations regarding relative consistency across parameter choices, bias due to counting self-matches, and dependence on sample length. SampEn was developed to combat these limitations[78,79]. It is defined as the negative logarithm of the conditional probability that two matching windows/embeddings of length $m$ still match when increasing their length to $m + 1$. A match between two windows is defined as having a distance smaller than $r$ times the standard deviation of the signal. Distance was calculated using the maximum norm and self-matches were excluded. Mathematically, it can be described as $SampEn(m, r) = -\log(\frac{A}{B})$ with $A$ being the number of matches with window size $m + 1$ and $B$ the number of matches of length $m$. Therefore SampEn measures how unlikely it is to find a window that matches another part of the signal where the continuation of the window is still a match. We set SampEn's parameters to $m = 2$ and $r = 0.2$ as suggested in the literature[78].

**Spectral entropy**. Complexity of the power spectrum or spectral entropy[80] (SpecEn) can be computed by applying entropy measures to the power spectrum of a signal. Traditionally, spectral entropy uses Shannon entropy[81], which disregards local patterns and only takes the signal's distribution of values into account. When applied to the power spectrum, Shannon entropy captures the amount of variance in the frequencies, which make up the EEG signal[82,83]. Additionally, we computed a variant of SpecEn, which we refer to as spectral sample entropy (SpecSampEn): Following the idea of SpecEn, we applied SampEn ($m = 2$, $r = 0.2$) to the full power spectrum. By looking at windows of multiple neighboring frequency bins, SampEn is able to incorporate local patterns in the power spectrum into its complexity estimate. This goes beyond the permutation invariant Shannon entropy and treats the power spectrum more as a coherent signal with meaningful relations between neighboring frequency bins, potentially capturing further aspects of complexity. The power spectrum we used to compute SpecEn and SpecSampEn was estimated as described in Section Feature extraction (Power spectral density), however, it was neither corrected for the $1/f$-like component nor split into frequency bands.

Note that by applying entropy not only to the raw EEG signal (SampEn), but also to the power spectrum (SpecEn), we gain insights into the degree of randomness in both the temporal and spectral domains of brain activity, providing complementary perspectives on neural dynamics.

**Lempel-Ziv complexity**. Lempel-Ziv complexity (LZc)[84] is a complexity measure designed for binary sequences and text. It counts the number of unique substrings in a sequence, thereby measuring how repetitive a signal is. That is, a less complex signal, according to LZc, consists of repetitions of a few different sub-strings while more complex signals are made up of non-repeating segments. As the initially proposed version of LZc is strongly influenced by signal length, we used a normalized variant, which scales LZc by $\frac{\log_b(n)}{n}$ where $n$ is signal length and $b$ the number of unique characters in the signal[85]. In line with standard practice, we applied a median split to the preprocessed EEG epochs to transform the signal into a binary sequence, allowing it to be processed by the Lempel-Ziv algorithm. In the context of neuroscience, LZc has been useful to track disorders of consciousness and was shown to reach a peak at the edge of chaos critical point[35].

**Detrended fluctuation analysis**. To measure the signal's long-term statistical dependence, we performed Detrended Fluctuation Analysis (DFA)[86,87] on the raw EEG signal using the AntroPy toolbox[88]. DFA, an estimate of self-affinity, divides the signal into windows and computes standard deviations as a function of window sizes. A power-law with exponent/slope $\alpha$ is then fitted to the resulting graph. The estimated $\alpha$ parameter is a generalization of the Hurst exponent, capable of dealing with non-stationary time series. Similar to the Hurst exponent, a slope less than 0.5 corresponds to an anti-correlated process, a slope of 0.5 indicates a white-noise process and a slope larger 0.5 suggests a correlated signal. In contrast to the Hurst exponent however, DFA can return slopes larger than one in the case of unbounded, non-stationary signals. Furthermore, the presence of long-range temporal correlations (LRTC) measured by DFA has been linked to systems operating near criticality[35,38,89].

We also analyzed the DFA scaling exponents of canonical frequency bands after computing the Hilbert transform. The frequency bands used were the same as in the PSD analysis, and we additionally calculated a broadband DFA in the range of 3 to 32 Hz. While our main focus is on DFA computed from the raw signal, these additional results are provided in Supplementary Fig. S4.

**Aperiodic slope**. We used the FOOOF algorithm[76] to estimate the slope of the aperiodic component of the EEG signal, which corresponds to the exponent in the $1/f$-like distribution of the signals's power spectrum. FOOOF was fit on the power spectrum between 3 and 32 Hz, limiting the number of periodic peaks to 5 and setting peak width limits to 0.5 and 12. The power spectrum was computed in the same way as described in Section Feature extraction (Power spectral density). FOOOF extracts the aperiodic slope from the power spectrum by fitting a parameterized model of the 1/f-like aperiodic component and separating it from the periodic oscillatory peaks. The aperiodic slope has previously been shown to reflect changes in E/I balance[57,60,61], which could link the effect of caffeine, an antagonist of the inhibitory neurotransmitter adenosine, to changes in E/I balance in the brain. Additionally, a flattening of the aperiodic slope is commonly seen as an indicator of a shift towards more critical dynamics[40–42]. For ease of interpretation we report the negative slope (i.e., positive values) in this work. A reduction in slope therefore, corresponds to a flatter distribution while an increased slope refers to a steeper decline in power.

**Direct statistical analysis**
To evaluate the difference between the caffeine and placebo conditions, two-sided paired permutation-based pseudo T-tests[90] were performed using exhaustive permutations ($n = 10^4$) and corrected for multiple comparisons using the maximum statistics method[91]. The T-tests were carried out on all extracted features (PSD bands, SpecEn, SampEn, SpecSampEn, PermEn). Statistical significance between the caffeine and placebo condition was evaluated at $p < 0.05$ and $p < 0.01$, and we report summarized effect sizes according to Cohen's $d$.

We additionally analyzed age-related differences by comparing young (20–27 years) and middle-aged (41–58 years) subgroups using the same paired t-test approach between caffeine versus placebo effects within each age group. To examine age-related differences in caffeine response, we conducted independent t-tests comparing the caffeine-placebo difference between age groups (($\text{caffeine}_{young} - \text{placebo}_{young}$) vs. ($\text{caffeine}_{middle-aged} - \text{placebo}_{middle-aged}$)). Statistical significance was corrected for multiple comparisons via the FDR Benjamini/Hochberg method and evaluated at $p < 0.05$ and $p < 0.01$.

**Supervised machine learning analysis**
To improve robustness and interpretability of the statistical results, we additionally trained supervised machine learning classifiers to distinguish between samples from the caffeine or placebo condition.

**Single-feature classification**. To assess the impact of caffeine on specific features in single channels, different machine learning classifiers were trained on single-feature, binary classification between the placebo and caffeine conditions. Two different classification algorithms were used, namely support vector machine (SVM)[92–95] with an RBF kernel and linear discriminant analysis (LDA)[96,97]. Models were trained and evaluated separately for the different sleep stages.

Permutation tests[98,99] ($n = 1000$) were applied to the trained model to determine the statistical significance of the classifiers' accuracy scores. The results were corrected for multiple comparisons using the maximum statistics method. During each permutation, the model's average score across folds in a grouped tenfold cross-validation was used. Due to the low sample size, we decided to additionally fit an LDA classifier on single-epoch data (sample sizes NREM: 26,776 caffeine and 30,818 placebo, REM: 9448 caffeine and 9893 placebo), instead of subject-wise averages. Here, we chose to evaluate classifier performance using the balanced accuracy metric (BAcc) due to class imbalance. While accuracy is biased towards the majority class, BAcc was shown not to overestimate classifier performance even in cases of extreme class imbalance[100].

**Multi-feature classification**. In addition to the exploration on a single-channel and single-feature level, we chose to implement a broader multi-feature classifier that takes all features from all channels to build a single model. Spectral power was computed from the power spectrum after removal of the $1/f$-like component, since our previous analysis showed a better separation between the two conditions when using corrected spectral power. We were able to access the contributions of individual features and channels by examining feature importance. To achieve this, we used a random forest, which estimates feature importance by averaging the relative rank of each feature across the decision trees that make up the forest. To measure variance in feature importance, the training process was repeated 1000 times for each sleep stage, applying grid search with grouped sevenfold cross-validation inside a nested cross-validation, leaving out samples from five different subjects in each iteration. See Supplementary Table S1 for the selection of hyperparameters optimized during grid search. The overall model score was determined using samples from the left-out testing subjects.

**Statistics and reproducibility**
EEG data were acquired from 40 healthy adult participants who each completed both placebo and caffeine conditions. For statistical analysis, features were averaged within subjects across artifact-free 20s EEG epochs, yielding 80 paired samples per feature. Two-sided, paired permutation-based pseudo t-tests ($n = 10^4$) were used to assess condition differences, with correction for multiple comparisons via the maximum statistics method. Age-related subgroup comparisons were evaluated using independent $t$-tests and FDR correction. Classifier significance was assessed using grouped cross-validation and permutation testing ($n = 1000$). Several independent analyses, including statistical tests and supervised machine learning models, yielded converging results, supporting the robustness of our findings.

**Reporting summary**
Further information on research design is available in Nature Portfolio Reporting Summary linked to this article.

**Data availability**
EEG data cannot be shared as participant consent for public data sharing was not obtained at the time of collection.

**Code availability**
All analyses were done with Python 3.8, utilizing the software modules scikit-learn[101] for machine learning, MNE-Python[102] for visualization, and, together with statsmodels[103], statistical analysis. The SciPy stack[104] was used for efficient data structures, mathematical operations, and plotting

functionalities. A large share of the code was developed inside the Jupyter Notebook and IPython framework[105]. The machine learning analysis was enabled in part by support provided by the Digital Research Alliance of Canada (alliancecan.ca). All code used for analysis and visualization can be found at github.com/dav0dea/caffeine-sleep[106].

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

## Author contributions

P.T., M.A., and T.L. contributed to the development of software. P.T. and K.J. conducted the main data analysis. S.F. and J.C. were responsible for data acquisition. P.T. and K.J. drafted the manuscript and contributed to its revision and adaptation. All authors were involved in the conceptualization and design of the study and approved the final version of the manuscript.

## Competing interests

The authors declare no competing interests.
