## [Transparent Peer Review file · Communications Biology]

Caffeine induces age-dependent increases in brain complexity and criticality during sleep

Corresponding Author: Mr Philipp Thölke

Version 0:

Reviewer comments:

Reviewer #1

(Remarks to the Author)

This paper is well written and has potential to be of high quality if given a greater level of detail and rigour, in particular with regard to the presentation and justification of methodology, with some suggestions outlined below:

Introduction

Criticality and its relevance to neural systems needs to be introduced.

The significance of different types of entropy needs to be introduced

The link between criticality and complexity needs to be introduced beyond just a correlative link

There is also a lack of introduction to the central hypothesis - what exactly would it mean for criticality and complexity to be increased with caffeine?

There is nothing introduced regarding the significance of periodic vs aperiodic components and their significance to the brain

Results

Effect sizes should be reported alongside p values.

Discussion

There should be some discussion regarding the link between criticality and complexity, as these are the main two themes of the paper - rather than just being discussed as separate measured.

There should be a deeper exploration as to how the chosen entropy measures relate to cognitive function

A discussion regarding possible mechanisms underlying age-related changes would be helpful

explain how changes in the DFA scaling exponent and aperiodic slope relate to neural dynamics and brain function in a broader sense

discuss how caffeine might induce changes in entropy and criticality — physiological mechanisms, such as changes in neurotransmitter systems or network connectivity

moreover, discuss where these findings place the paper in broader context within computational neuroscience — how does this advance understanding of neural dynamics and is there potential for e.g. understanding other neurological conditions — future directions?

Methods

The authors explain how the complexity measures are calculated, but not how criticality is being quantified? The ways in which the DFA scaling exponent and the aperiodic slope of the power spectrum are calculated need to be detailed — what are the frequency ranges being considered — how is the slope being extracted from the log-log plot — and moreover, why is this a suitable indicator of criticality, given the limited size of the system

ensure that statistical methods used to compare caffeine vs. placebo are more thoroughly described — e.g. stat tests used, effect sizes — and a power analysis or comparison with sample sizes of previous studies would help justify the sample size

The authors should explain steps taken to validate their measures so as to ensure robustness — cross-validation, sensitivity analysis, comparison to other measures of criticality

An explanation of why the chosen entropy metrics were chosen over others would be helpful.

Reviewer #2

(Remarks to the Author)

Thölke et al. recorded sleep EEG with and without administration of 200 mg of caffeine and investigated the effects separately for non-REM and REM sleep using a wide range of spectral and complexity-related quantitative EEG biomarkers. To further illustrate the relative importance of the different EEG biomarkers, the authors applied machine learning to test the accuracy of discriminating between the caffeine and placebo conditions. Interestingly, NREM sleep seemed most affected by caffeine and this was most clearly reflected in the change of complexity measures compared to spectral measures.

Overall, I applaud the use of many different biomarker algorithms to dissect the effect of caffeine, which, as the authors point out, is the most widely consumed psychoactive stimulant. However, I feel the authors could do more to clarify the added value of using so many measures and condense the main story. Specifically:

1. I found the paper a bit heavy and technical to read for various reasons:

- a. Figure legends miss take-home messages. Right now, they are neutral and technical, and their meaning is only possible when consulting the main text.
- b. Similarly, I missed an active wording of 'shift in criticality' in terms of becoming more sub- or super-critical, let alone how that pairs with the 'boost in complexity'. This reviewer was surprised that increasing entropy (randomness, right?) was seen as a greater complexity. I would say criticality is complexity. I very much missed seeing signals with an explanation of what features in these signals were regarded "complex".
- c. The journal style of presenting Methods at the end left this reviewer feeling a lot of information was missing, such as an experimental paradigm and qualitative explanation of algorithms used. At a minimum, the authors could make cross-references to Methods at appropriate places; however, some explanations in the Intro and results would be more reader-friendly.
- d. Whereas the extensive use of algorithms has its appeal, I could also see why the remarkably similar effects of the entropy measures could be treated and presented differently, e.g., by stating that 'the measures were correlated and because of being strongly correlated and showing similar effects (Supplementary figure xx), we focus on yy'. If some of the numerous topographic maps were moved to the supplementary material, there would be more figure space for showing the values with the benefit of allowing the reader to evaluate the variability of these measures and the consistency of the effect of caffeine.

Minor comments:

1. I miss n-numbers on figures for easier readability, e.g., fig. 5.
2. I think it is not very convincing that there are age, considering the lower statistical power. Besides, just because a channel reaches significance in one group, but not another does not imply that there is a significant effect of age. Thus, I would either use statistics to formally test the effect of age or downplay the robustness of the differences and publish Fig. 5 as a supplementary figure. I certainly would not say that the data support the conclusion in the Discussion, "the brains of the middle-aged individuals were strikingly less perturbed by caffeine than those of the younger subjects," and suggest not to highlight the age effect in the title.
3. The conclusion in the last sentence of the abstract is too vague. Write the novel insight and what implications this has instead of leaving this up to the reader.
4. I don't understand why the 1st column in Fig. 1 differs from the 1st column of Fig. 2.
5. If I understand correctly, the authors applied DFA to the "raw" bandpass filtered data as opposed to the amplitude envelope used in the literature referenced in Hardstone et al. 2012. That is very surprising, and the DFA exponents are also usually high. Considering that the DFA was used in such an untraditional fashion, it would seem natural to show the grand average fluctuation functions that the DFA exponents were fit to. The time scales of the authors fit the DFA are also not reported.

Reviewer #3

(Remarks to the Author)

Thoelke et al. reanalyze an EEG sleep dataset (N=40) of participants under prior caffeine consumption vs. placebo. Analyses are performed separately for REM and NREM epochs of 20 seconds duration. They report that caffeine reduces

low frequency power in NREM sleep and, alongside, leads to an increase in several complexity measures. The study recovers well known effects of caffeine on sleep electrophysiology, some of which is cited by the authors.

- My main concern is with the novelty of the reported findings. The authors report a reduction of low frequency power in NREM with caffeine, which is well known and a general sign of less slow wave sleep (SWS). It has been shown in several studies that SWS reduces EEG complexity. With reduction of SWS under caffeine it is thus expected that complexity (e.g. measured by entropy, DFA, ...) is relatively increased. Thus, I am hesitant to see the novelty this study provides as all reported effects are essentially only a consequence of the reduced SWS induced by caffeine, which is a well known effect.
- There is also some work on using ML to classify epochs under caffeine from controls using these features (power, complexity etc.)
- Minor: Scoring sleep stages: the authors use S3, S4 stages -> now according to new classification just S3 (or N3) should be used. Since the authors combine all NREM stages into one class, this does not matter too much.
- Minor: Fig.2 b, please add units to axes. Is the y-axis in log-scale?

Version 1:

Reviewer comments:

Reviewer #1

(Remarks to the Author)

The updated manuscript is significantly improved, particularly with regard to methodology. Some minor improvements could still be made regarding the points below:

- 1) When it comes to the age related effects, some interpretation of the differences between younger and middle aged participants is needed. For instance, how might baseline differences in sleep architecture or receptor density influence the observed effects? Could age-related cognitive or lifestyle factors play a role?
- 2) There is a strong focus on healthy individuals - but how might the findings be generalized to larger populations with sleep disorders or neurodegenerative conditions?
- 3) the potential caffeine induced shifts in nrem architecture is given as a limitation - how might such shifts influence complexity and criticality?

Reviewer #2

(Remarks to the Author)

The authors have addressed my previous comments and suggestions well.

Ms. No.: COMMSBIO-24-3851 (Previous submission)

Title: Caffeine induces age-dependent increases in brain complexity and criticality during sleep

In the following, we provide point-by-point responses to all the comments and questions raised by Reviewers #1, #2 and #3, from a previous submission of this study to Nature Communications Biology (COMMSBIO-24-3851). For the sake of clarity, the Reviewers' comments are in italic text, our responses are in bold font, and additions/changes made to the manuscript are presented here (as well as in the revised manuscript) in blue.

Reviewer #1:

This paper is well written and has potential to be of high quality if given a greater level of detail and rigour, in particular with regard to the presentation and justification of methodology, with some suggestions outlined below:

Introduction

Criticality and its relevance to neural systems needs to be introduced.

We have expanded the introduction to clarify the relevance of brain criticality. Specifically, we describe how criticality theory highlights an optimal balance between order and randomness, enabling efficient information processing and cognitive flexibility. We also discuss how metrics such as Lempel-Ziv complexity, power spectrum slope, and long-range temporal correlations can capture criticality-related neural dynamics. Furthermore, we address how aging affects these markers, providing context for understanding caffeine's potential modulatory effects.

[...] Type 2 complexity addresses this limitation by highlighting that maximal complexity arises in systems that achieve an optimal balance between order and randomness. According to criticality theory, this critical point characterizes the state of maximal computational efficiency and optimal information processing. Such states, often associated with criticality, enable the brain to balance stability and flexibility, a hallmark of higher-order cognitive processes [1-3]. As a matter of fact, the so called edge of chaos criticality, characterized by a balance between order and disorder, can be assessed through various metrics including Lempel-Ziv complexity, the slope of the aperiodic power spectrum, and long-range temporal correlations [1,4]. These measures have been shown to track cognitive states and arousal levels, suggesting they capture functionally relevant aspects of neural dynamics.

The significance of different types of entropy needs to be introduced

We agree with the reviewer. We have now clarified the distinction between the types of entropy used in this work. By applying entropy to both the raw EEG signal and the power spectrum, we capture randomness in the temporal and spectral domains, offering complementary insights into neural dynamics.

Note that by applying entropy not only to the raw EEG signal (SampEn), but also to the power spectrum (SpecEn), we gain insights into the degree of randomness in both the temporal and spectral domains of brain activity, providing complementary perspectives on neural dynamics.

The link between criticality and complexity needs to be introduced beyond just a correlative link

We have clarified the distinction between Type 1 and Type 2 complexity and their relationship to criticality. While Type 1 complexity increases linearly with randomness, Type 2 complexity follows an inverted U-shape, peaking at an optimal balance between order and randomness. Criticality theory aligns with Type 2 complexity, describing a state of maximal computational efficiency and optimal information processing, enabling the brain to balance stability and flexibility—key characteristics of higher-order cognitive processes.

The complexity of a system can generally be divided into two main subtypes: Type 1 complexity increases linearly with randomness (e.g. entropy or Lempel-Ziv complexity), while Type 2 complexity follows an inversely parabolic pattern, peaking at moderate randomness and declining at extremes (e.g. criticality or Kolmogorov Complexity) [5,6].

[...] High entropy alone does not necessarily indicate maximal complexity in brain function. In contrast, Type 2 complexity addresses this limitation by highlighting that maximal complexity arises in systems that achieve an optimal balance between order and randomness. According to criticality theory, this critical point characterizes the state of maximal computational efficiency and optimal information processing. Such states, often associated with criticality, enable the brain to balance stability and flexibility, a hallmark of higher-order cognitive processes [1-3].

There is also a lack of introduction to the central hypothesis - what exactly would it mean for criticality and complexity to be increased with caffeine?

We have expanded our introduction to establish clear hypotheses linking caffeine's effects to brain complexity and criticality. Our primary hypothesis posits that by blocking adenosine and altering sleep architecture, caffeine increases EEG

complexity and shifts brain dynamics toward criticality during NREM sleep. This prediction builds on established links between cognitive performance, brain complexity, and criticality, as well as known effects of caffeine on sleep architecture. We also hypothesize that these effects are attenuated in middle-aged individuals due to age-related changes in adenosine receptor density and baseline neural dynamics. The revised introduction now provides a stronger theoretical framework connecting sleep, caffeine, aging, and brain dynamics.

In wakefulness, caffeine ingestion facilitates alertness and cognitive performance by blocking the action of adenosine, a neurotransmitter that promotes sleep drive. Given that cognitive performance is closely linked to brain complexity and criticality [3], one would expect that caffeine intake leads to an increase in EEG complexity or entropy and a shift closer to a critical regime. In fact, a previous fMRI study has provided evidence for caffeine-induced increases in brain entropy during wakefulness [7]. However, it is yet unknown whether similar effects extend to sleep EEG signals.

Furthermore, caffeine is known to alter sleep architecture by reducing slow-wave sleep (SWS) and increasing lighter sleep stages such as N1 and N2 [8,9]. Complexity measures, including entropy, have been shown to reliably reflect these changes, with entropy being highest in wakefulness, followed by REM sleep, and progressively decreasing across N1, N2, and SWS [10,11]. We expect that caffeine-induced alterations in sleep architecture, specifically the reduction of SWS and the increase in lighter sleep stages (N1 and N2), will result in a measurable increase in EEG complexity during NREM sleep.

Aging is characterized by a decrease in adenosine receptor density, reduced time spent in deep sleep, and shifts in brain dynamics, including increased neural entropy and a flattened power spectrum slope [11-14]. Given these baseline changes and caffeine's known effects on arousal and sleep architecture, we anticipate that caffeine's impact on brain complexity and criticality during sleep will be weaker in middle-aged compared to younger individuals.

Based on the above, the main hypothesis of this study is that caffeine ingestion leads to increased EEG complexity and a shift closer to a critical regime during NREM sleep. Our secondary hypothesis is that caffeine-induced changes in brain complexity and criticality are weaker in middle-aged compared to younger individuals, reflecting known age-related differences in adenosine receptor density, sleep architecture, and neural dynamics.

There is nothing introduced regarding the significance of periodic vs aperiodic components and their significance to the brain

We agree that this would be useful in the introduction. We have now introduced the significance of periodic and aperiodic components in EEG activity. In a nutshell, we explain that periodic activity reflects oscillatory synchronization, while aperiodic activity provides insights into the scale-invariant properties of the signal, and is thought to capture the balance between neural excitation and inhibition. This distinction is crucial for understanding caffeine's effects on brain dynamics, as changes in the aperiodic component may indicate shifts in cortical excitability, while

periodic changes can highlight alterations in specific oscillatory rhythms essential for sleep and cognitive processes.

Disentangling periodic and aperiodic components in the EEG spectra enables the isolation of distinct neural processes, with periodic activity reflecting oscillatory synchronization and aperiodic activity providing insights into scale-invariant properties of the EEG and the 1/f-behavior of its power spectrum, which has been linked to the neural excitation and inhibition balance [15,16]. This distinction is particularly relevant for understanding how caffeine modulates brain dynamics, as changes in the aperiodic component may highlight shifts in cortical excitability, while periodic changes can reveal alterations in specific oscillatory rhythms critical for sleep and cognitive function.

Results

Effect sizes should be reported alongside p values.

Thanks for this suggestion. We have included effect sizes (Cohen's d) alongside p-values in the manuscript and provided a supplementary table summarizing the largest effect sizes for key statistical findings across sleep stages and features.

Feature	NREM	REM
delta	-0.633**	0.421
theta	-0.504*	-0.637**
alpha	-0.641**	0.306
sigma	-0.264	-0.296
beta	0.641**	-0.512*
SpecEn	0.668**	0.603**
SampEn	0.696**	0.578**
SpecSampEn	0.933**	0.402
LZc	0.712**	0.589**
scaling exponent	-0.668**	-0.531*
aperiodic slope	-0.612**	-0.512*

Table S3. Peak effect sizes across features. Cohen's d values with largest absolute values for each feature and sleep stage. Significance levels: $p < 0.05$ (*) and $p < 0.01$ (**).

Discussion

There should be some discussion regarding the link between criticality and complexity, as these are the main two themes of the paper - rather than just being discussed as separate measured.

We have expanded on the connection between complexity and criticality, highlighting how changes in entropy (Type 1 complexity) align with shifts in criticality (Type 2 complexity). The caffeine-induced increase in sample entropy (SampEn) during NREM

sleep suggests enhanced information processing, consistent with states near the critical point characterized by maximal integration and optimal information flow across scales. Similarly, the observed increase in Lempel-Ziv complexity (LZc), which reflects the richness of neural information content, supports this interpretation. LZc has been shown to inversely track the distance to edge-of-chaos criticality [1], indicating that caffeine may shift brain activity closer to the critical regime, particularly during NREM sleep.

In a prior investigation assessing the complexity of biological systems, low approximate entropy (ApEn) was associated with the isolation and segregation of dynamical processes [17]. Furthermore, experimental work suggests increased segregation of the brain during NREM sleep [18,19], which was later shown to be reflected in decreased ApEn [20]. These observations provide an interesting framework in which to interpret the caffeine-induced increase in sample entropy (SampEn) which we found during NREM sleep. Given that SampEn is closely related to ApEn, our results may therefore indicate that caffeine ingestion reduces the separation between brain networks, promoting greater integration. These results also suggest an increase in information processing during NREM, which brain entropy is commonly seen as a proxy for [21]. This relationship between entropy and information processing aligns with theories of criticality in neural systems, where states near the critical point are characterized by maximal integration across scales and optimal information processing capacity [3]. The higher SampEn values we observed after caffeine administration may indicate a shift away from the subcritical regime, where local processing dominates and in which the brain operates during NREM sleep [22]. This shift towards the critical point may reflect a transition into a more integrated state that allows for enhanced information flow across multiple temporal scales, which also aligns with our findings on LRTC, as evaluated by the DFA scaling exponent.

Given that Lempel-Ziv complexity (LZc) quantifies the compressibility of neural signals – another measure of the richness of information content – it is not surprising to see that caffeine's influence on LZc and entropy metrics was similar, displaying widespread increases during NREM sleep and a localized increase in occipital channels during REM sleep compared to placebo. Interestingly, LZc has previously been shown to inversely track the distance to criticality in the brain, i.e. an increase in LZc is tied to activity closer to the critical point [1], which further supports the hypothesis that caffeine causes a shift from subcritical dynamics closer to the critical point, particularly during NREM sleep.

There should be a deeper exploration as to how the chosen entropy measures relate to cognitive function

In the revised manuscript we go into more details regarding the relationship between the chosen entropy measures and cognitive function. Higher brain entropy during sleep is associated with enhanced information integration and dynamic adaptability, processes essential for sleep-dependent cognitive functions such as memory consolidation and synaptic plasticity. The caffeine-induced increase in EEG entropy, particularly during NREM sleep—a stage typically characterized by lower

entropy—suggests potential alterations in these processes. This increase may reflect disrupted neural communication and reduced efficiency in sleep-related cognitive functions. While heightened entropy during sleep has been linked to conditions such as hypertension and early-stage Alzheimer’s disease, the cognitive consequences of pharmacologically induced changes in sleep brain entropy warrant further investigation.

Higher entropy in neural signals during sleep has been associated with enhanced information integration and dynamic adaptability, processes that are critical for cognitive functions reliant on sleep, such as optimal neural communication and adaptive information processing. The caffeine-induced increase in EEG entropy during sleep may therefore reflect changes to these processes, potentially impacting the brain's ability to efficiently process and integrate information across different neural states. As the caffeine-induced increase of brain entropy was most pronounced in NREM, the stage known to exhibit low brain entropy, it is tempting to associate the effect of caffeine with a deterioration of sleep quality. Although increased brain entropy during sleep has been linked to hypertension [23] and early-stage Alzheimer’s disease [24], further investigation is needed to elucidate the impact of pharmacologically induced alterations in sleep brain entropy and their implications for sleep-dependent cognitive functions.

A discussion regarding possible mechanisms underlying age-related changes would be helpful

We agree with this (and other reviewers) that the age-dependent observations needed more interpretation. In the new version of the article, we highlight the age-dependent changes in terms of (i) Adenosine activity, (ii) sleep structure, and (iii) EEG aperiodic component.

[...] Adenosine is a key regulator of sleep-wake dynamics, and its effects are mediated primarily through A_1 and A_{2a} adenosine receptors [25]. Aging is associated with a natural decline in A_1 receptor density [26], which likely reduces the capacity of adenosine to modulate sleep-related processes, particularly during REM sleep, where adenosine activity is already lower compared to NREM sleep [25].

In younger adults, higher A_1 receptor concentrations may amplify the effects of caffeine, which acts by blocking adenosine binding to these receptors. During REM sleep, this interaction could result in a compound effect of reduced adenosine activity (characteristic of REM) and higher receptor availability, enabling caffeine to exert a more pronounced influence on brain dynamics. By contrast, the diminished A_1 receptor density in middle-aged adults likely limits caffeine’s impact, as fewer receptors are available for adenosine binding and subsequent blockade by caffeine.

The absence of significant age effects during NREM sleep may also be explained by the interplay of these factors. Adenosine activity is upregulated during NREM sleep [25], and this robust baseline activity may mask age-related differences in receptor availability. Consequently, while caffeine induces comparable changes in NREM-related EEG features

across age groups, the distinct dynamics of adenosine and receptor density during REM sleep appear to drive the observed age-dependent effects.

explain how changes in the DFA scaling exponent and aperiodic slope relate to neural dynamics and brain function in a broader sense

We have now clarified how changes in the DFA scaling exponent and aperiodic slope relate to neural dynamics, particularly in terms of excitation-inhibition balance and brain criticality. Caffeine-induced flattening of the 1/f slope and a decrease in the scaling exponent suggest a shift towards increased excitation. These changes align with models linking the 1/f slope and related exponents to the excitation-inhibition ratio. This shift could push the brain's dynamics closer to a critical state, which is thought to optimize computational efficiency. Our findings indicate that caffeine reduces inhibition-dominated dynamics during sleep, especially in NREM, promoting a state of higher excitation.

Both modeling and experimental studies have shown that the slope of the 1/f component of the power spectrum (and related measures like scaling and Hurst exponents) can serve as indicators of the excitation-to-inhibition (E:I) ratio [15,16,27-30]. Our data reveal that caffeine induces a flattening of the 1/f slope and a drop in the scaling exponent, providing evidence for a shift towards increased excitation. Interestingly, this finding takes on additional significance within the framework of brain criticality, where E:I balance serves as a control parameter that can drive neural dynamics closer to or further from the critical point. The critical point represents a state poised between order and chaos that allows for maximal computational efficiency and flexibility [3]. Our results suggest that caffeine reduces the characteristic inhibition-dominated dynamics typically observed during sleep, particularly during NREM sleep, shifting the system towards a state of increased excitation.

discuss how caffeine might induce changes in entropy and criticality — physiological mechanisms, such as changes in neurotransmitter systems or network connectivity

We expanded on how caffeine induces changes in entropy and brain criticality, linking these changes to neurotransmitter systems and network connectivity. Caffeine reduces adenosine binding, which not only affects adenosinergic inhibition but also triggers cascading effects across multiple neurotransmitter systems, including increased dopamine and norepinephrine, and enhanced acetylcholine availability. These changes alter the excitation-inhibition balance, which influences brain dynamics and criticality. Specifically, our data show that caffeine flattens the 1/f slope and reduces the scaling exponent, signaling a shift toward increased excitation, especially during NREM sleep. This shift may move the brain's dynamics closer to a critical state, promoting greater computational efficiency. Additionally, caffeine's effects on entropy, reflected by increased sample entropy (SampEn) during NREM, suggest reduced isolation and greater system flexibility.

While adenosinergic inhibition mostly targets excitatory neurons (specifically acetylcholine and glutamate), it also reduces GABAergic activity [31], a key modulator of beta oscillations

in the brain [32-34]. Taken together, the caffeine-induced increase in beta power might be directly attributable to elevated GABA levels, which result from reduced adenosinergic inhibition of GABAergic neurons.

[...] caffeine not only affects adenosine signaling directly but also triggers a cascade of changes in other neurotransmitter systems, including increased dopamine and norepinephrine release, enhanced acetylcholine availability, and modulation of the balance between GABAergic and glutamatergic transmission [31-35]. While these complex interactions across multiple neurotransmitter systems make it challenging to predict caffeine's global effects on brain dynamics, examining changes in the excitation-inhibition balance may provide a more tractable framework for understanding its impact.

Our data reveal that caffeine induces a flattening of the $1/f$ slope and a drop in the scaling exponent, providing evidence for a shift towards increased excitation. Interestingly, this finding takes on additional significance within the framework of brain criticality, where E:I balance serves as a control parameter that can drive neural dynamics closer to or further from the critical point. The critical point represents a state poised between order and chaos that allows for maximal computational efficiency and flexibility [3]. Our results suggest that caffeine reduces the characteristic inhibition-dominated dynamics typically observed during sleep, particularly during NREM sleep, shifting the system towards a state of increased excitation.

In a prior investigation assessing the complexity of biological systems, low approximate entropy (ApEn) was associated with the isolation and segregation of dynamical processes [17]. Furthermore, experimental work suggests increased segregation of the brain during NREM sleep [18,19], which was later shown to be reflected in decreased ApEn [20]. These observations provide an interesting framework in which to interpret the caffeine-induced increase in sample entropy (SampEn) which we found during NREM sleep. Given that SampEn is closely related to ApEn, our results may therefore indicate that caffeine ingestion reduces the separation between brain networks, promoting greater integration.

moreover, discuss where these findings place the paper in broader context within computational neuroscience — how does this advance understanding of neural dynamics and is there potential for e.g. understanding other neurological conditions — future directions?

The caffeine-induced shift in criticality and increase in EEG entropy during sleep suggest alterations in how the brain processes and integrates information across neural states. These findings provide a new perspective on the effects of caffeine on sleep, with implications for understanding brain entropy in neurological conditions like hypertension and Alzheimer's disease. Future research should further explore the

impact of caffeine on sleep-dependent cognitive functions and consider how its effects on the excitation-inhibition balance could inform treatments for neurological conditions that involve disrupted criticality. Additionally, investigating the influence of caffeine on individual NREM stages and sleep architecture will be important for isolating its specific effects on neural dynamics during sleep.

In addition, discarding the aperiodic component uncovered a strong statistically significant increase in beta power over multiple parietal, central and frontal sites. While adenosinergic inhibition mostly targets excitatory neurons (specifically acetylcholine and glutamate), it also reduces GABAergic activity [31], a key modulator of beta oscillations in the brain [32-34]. Taken together, the caffeine-induced increase in beta power might be directly attributable to elevated GABA levels, which result from reduced adenosinergic inhibition of GABAergic neurons.

The caffeine-induced increase in EEG entropy during sleep may therefore reflect changes to these processes, potentially impacting the brain's ability to efficiently process and integrate information across different neural states. As the caffeine-induced increase of brain entropy was most pronounced in NREM, the stage known to exhibit low brain entropy, it is tempting to associate the effect of caffeine with a deterioration of sleep quality. Although increased brain entropy during sleep has been linked to hypertension [23] and early-stage Alzheimer's disease [24], further investigation is needed to elucidate the impact of pharmacologically induced alterations in sleep brain entropy and their implications for sleep-dependent cognitive functions.

Interestingly, the link we make here between caffeine's antagonistic effect on adenosine receptors and the observed shift in criticality is conceptually consistent with previous research showing that by blocking GABA receptors and thus reducing inhibitory synaptic transmission, an artificial upward shift in excitation can occur, leading to a supercritical state with larger-than-expected neuronal avalanches [36,37]. This suggests that, while adenosine blockers have an upregulating effect on GABAergic neurons, the global impact of caffeine on the excitation-inhibition (E:I) balance appears to be positive, as indicated in the literature [31].

[...] caffeine is known to alter the ratio of NREM stages (N1, N2, SWS) to each other, making it difficult to disentangle the effect of caffeine on individual NREM stages from the shift in sleep architecture from deeper to lighter sleep. Many of the caffeine-related significant changes we found in NREM also occur during the shift from deeper to lighter sleep. Thus, further work should look into this.

Methods

The authors explain how the complexity measures are calculated, but not how criticality is being quantified? The ways in which the DFA scaling exponent and the aperiodic slope of the power spectrum are calculated need to be detailed — what are the frequency ranges being considered — how is the slope being extracted from the log-log plot — and moreover, why is this a suitable indicator of criticality, given the limited size of the system

We have clarified the methodology used to quantify criticality and calculate the DFA scaling exponent and aperiodic slope. Criticality is assessed using measures such as LZc and DFA, which are linked to systems operating near critical points, and the flattening of the aperiodic slope is commonly seen as indicative of critical dynamics. DFA was performed on 20s EEG epochs using the AntroPy toolbox, while the aperiodic slope was extracted with the FOOOF algorithm, considering the power spectrum between 3 and 32 Hz. Despite the finite size of the system, these measures are valid indicators of critical dynamics, as finite systems can still exhibit scale invariance and power-law relationships over a range of scales.

How criticality is quantified:

In the context of neuroscience, LZc has been useful to track disorders of consciousness (DOC) and was shown to reach a peak at the edge of chaos critical point [1].

Furthermore, the presence of long range temporal correlations (LRTC) measured by DFA has been linked to systems operating near criticality [1,4,38].

Additionally, a flattening of the aperiodic slope is commonly seen as an indicator of a shift towards more critical dynamics [12-14].

How DFA scaling exponent and aperiodic slope are calculated:

For the sake of consistency, all features were extracted from 20s epochs of continuous EEG.

[...] we performed Detrended Fluctuation Analysis (DFA) [39,40] on the raw EEG signal using the AntroPy toolbox [41].

We used the FOOOF algorithm [42] to estimate the slope of the aperiodic component of the EEG signal [...]. FOOOF was fit on the power spectrum between 3 and 32 Hz, limiting the number of periodic peaks to 5 and setting peak width limits to 0.5 and 12. The power spectrum was computed in the same way as described in Section 3.5 (Power spectral density). FOOOF extracts the aperiodic slope from the power spectrum by fitting a

parameterized model of the 1/f-like aperiodic component and separating it from the periodic oscillatory peaks.

Suitability as an indicator of criticality:

Note that while the critical point is formally defined only for infinite systems, finite systems can still operate in a critical regime, exhibiting approximate scale invariance and power-law relationships over a range of scales. Therefore, the criticality features discussed below, such as DFA scaling exponent and aperiodic slope, remain valid indicators of critical dynamics even in finite neural systems.

ensure that statistical methods used to compare caffeine vs. placebo are more thoroughly described — e.g. stat tests used, effect sizes — and a power analysis or comparison with sample sizes of previous studies would help justify the sample size

We have expanded the statistical methodology section to provide a comprehensive description of our analyses. We detail the use of two-sided paired permutation-based pseudo T-tests with exhaustive permutations, including correction for multiple comparisons. Effect sizes are now reported using Cohen's d. The analysis includes separate evaluations for age-specific subgroups, with additional tests to examine age as a confounding variable. For the machine learning components, we implemented permutation tests to validate the statistical significance of classifier performance.

To evaluate the difference between the caffeine and placebo conditions, two-sided paired permutation-based pseudo T-tests [43] were performed using exhaustive permutations ($n=10^4$) and corrected for multiple comparisons using the maximum statistics method [44]. The T-tests were carried out on all extracted features (PSD bands, SpecEn, SampEn, SpecSampEn, PermEn). Statistical significance between the caffeine and placebo condition was evaluated at $p<0.05$ and $p<0.01$, and we report summarized effect sizes according to Cohen's d.

We additionally analyzed age-related differences by comparing young (20-27 years) and middle-aged (41-58 years) subgroups using the same paired t-test approach between caffeine versus placebo effects within each age group. To examine age-related differences in caffeine response, we conducted independent t-tests comparing the caffeine-placebo difference between age groups ($(\text{caffeine}_{\text{young}} - \text{placebo}_{\text{young}})$ vs. $(\text{caffeine}_{\text{middle-aged}} - \text{placebo}_{\text{middle-aged}})$). Statistical significance was corrected for multiple comparisons via the FDR Benjamini/Hochberg method and evaluated at $p<0.05$ and $p<0.01$.

Permutation tests [45,46] ($n=1000$) were applied to the trained model to determine the statistical significance of the classifiers' accuracy scores. The results were corrected for multiple comparisons using the maximum statistics method.

The authors should explain steps taken to validate their measures so as to ensure robustness — cross-validation, sensitivity analysis, comparison to other measures of criticality

We employed multiple validation approaches to ensure robustness. Our cross-validation strategy included grouped 10-fold validation for subject-level analysis and nested cross-validation with grouped 7-fold validation for hyperparameter optimization. To address sample size limitations, we conducted additional analysis using single-epoch data, significantly increasing our sample size. We used balanced accuracy metrics to account for class imbalance, as this measure remains reliable even with uneven class distributions.

During each permutation, the model's average score across folds in a grouped 10-fold cross-validation was used. Due to the low sample size we decided to additionally fit an LDA classifier on single-epoch data (sample sizes NREM: 26,776 caffeine and 30,818 placebo, REM: 9448 caffeine and 9893 placebo), instead of subject-wise averages. Here, we chose to evaluate classifier performance using the balanced accuracy metric (BAcc) due to class imbalance. While accuracy is biased towards the majority class, BAcc was shown not to overestimate classifier performance even in cases of extreme class imbalance [47].

[...] applying grid search with grouped 7-fold cross-validation inside a nested cross-validation, leaving out samples from five different subjects in each iteration.

An explanation of why the chosen entropy metrics were chosen over others would be helpful.

We selected these entropy metrics based on their established reliability and complementary nature. While Sample Entropy quantifies temporal complexity in the EEG signal, Spectral Entropy measures the disorder in frequency distributions. This dual approach provides insight into both temporal and spectral aspects of neural dynamics under different cognitive states.

Note that by applying entropy not only to the raw EEG signal (SampEn), but also to the power spectrum (SpecEn), we gain insights into the degree of randomness in both the temporal and spectral domains of brain activity, providing complementary perspectives on neural dynamics.

We thank Reviewer 1 for the detailed review and all the helpful and on-point comments. We believe the manuscript has substantially improved as a result of the changes and additions suggested by this reviewer.

Reviewer #2:

Thölke et al. recorded sleep EEG with and without administration of 200 mg of caffeine and investigated the effects separately for non-REM and REM sleep using a wide range of spectral and complexity-related quantitative EEG biomarkers. To further illustrate the relative importance of the different EEG biomarkers, the authors applied machine learning to test the accuracy of discriminating between the caffeine and placebo conditions. Interestingly, NREM sleep seemed most affected by caffeine and this was most clearly reflected in the change of complexity measures compared to spectral measures.

Overall, I applaud the use of many different biomarker algorithms to dissect the effect of caffeine, which, as the authors point out, is the most widely consumed psychoactive stimulant. However, I feel the authors could do more to clarify the added value of using so many measures and condense the main story. Specifically:

1. I found the paper a bit heavy and technical to read for various reasons:

a. Figure legends miss take-home messages. Right now, they are neutral and technical, and their meaning is only possible when consulting the main text.

We have updated all figure legends to clearly convey their key findings. Each legend now explains the main patterns shown in the data, the directionality of effects, and their statistical significance. The revised legends help readers grasp the central message of each figure without needing to consult the main text, while maintaining necessary technical details for proper interpretation of the visualization methods used.

Fig 1. Brain activity patterns during sleep (NREM and REM), comparing caffeine versus placebo effects on periodic neural oscillations (after removing aperiodic spectral components). Left column shows statistical differences (blue: reduced during caffeine, red: increased during caffeine), while SVM/LDA columns show classification accuracy between conditions (green). Dots indicate statistical significance (grey: $p < 0.05$, white: $p < 0.01$).

Fig 2. Assessing the impact of changes in the slope of the aperiodic component on spectral power in canonical frequency bands. **(a)** Brain activity before (uncorrected) and after (corrected) removing the aperiodic (1/f) component from the power spectrum. The topographic maps show t-values (caffeine - placebo condition) with blue indicating reduced and red increased spectral power during caffeine (dots: grey $p < 0.05$, white $p < 0.01$). **(b)** Illustrative example of caffeine-induced shifts in aperiodic slope (solid lines), showing full power spectra (dashed lines) from a single subject at electrode Fz (red star) in a log-log plot. Subject and channel were chosen to be representative for the effect found across subjects. This panel does not contain results of statistical tests but rather serves as a clarification to the reader about the effect of the aperiodic slope on power spectral density across frequency bands.

Fig 3. Comparison of caffeine versus placebo effects on brain complexity and criticality measures during NREM and REM sleep. Left columns show statistical differences (blue: reduced during caffeine, red: increased during caffeine). Middle and right columns show classification accuracy (green) from SVM and LDA models, validated using permutation tests. Dots indicate statistical significance (grey: $p < 0.05$, white $p < 0.01$). Most prominently we see broad caffeine-induced increases in entropy and complexity, and a flattening of the aperiodic slope.

Fig 4. Feature importance across brain regions during NREM and REM sleep, derived from random forest models trained on 220 features (11 features \times 20 channels). Bar plots rank input dimensions by feature importance, with warm colors (red to yellow) showing spectral power bands and cold colors (blue and green) showing measures related to entropy, complexity and criticality. Topographic maps display the spatial distribution of importance values, averaged across 1000 models per sleep stage. Darker colors indicate higher feature importance.

Fig 5. Topographical maps showing age-related differences in brain responses to caffeine versus placebo. The "young" (20-27 years) and "middle-aged" (41-58 years) columns display t-values of paired T-tests between caffeine vs. placebo. To rule out statistical effects of sample size, the remaining columns show the results of independent T-tests (t- and p-values) between age groups ((caffeine_{young} - placebo_{young}) vs. (caffeine_{middle-aged} - placebo_{middle-aged})). Grey dots indicate $p < 0.05$, white dots $p < 0.01$.

b. Similarly, I missed an active wording of 'shift in criticality' in terms of becoming more sub- or super-critical, let alone how that pairs with the 'boost in complexity'. This reviewer was surprised that increasing entropy (randomness, right?) was seen as a greater complexity. I would say criticality is complexity. I very much missed seeing signals with an explanation of what features in these signals were regarded "complex".

We thank the reviewer for pointing this out. We have now clarified the relationship between different types of complexity in our analysis. Type 1 complexity measures like entropy quantify randomness and signal unpredictability, while Type 2 complexity measures like criticality capture the balance between order and disorder. We explain how increased entropy during caffeine administration reflects heightened neural activity and information processing, while changes in criticality indicate shifts in the brain's dynamic organization. This distinction helps interpret our findings within the broader theoretical framework of neural complexity. The manuscript now includes relevant citations that ground these concepts in established literature.

The complexity of a system can generally be divided into two main subtypes: Type 1 complexity increases linearly with randomness (e.g. entropy or Lempel-Ziv complexity), while Type 2 complexity follows an inversely parabolic pattern, peaking at moderate randomness and declining at extremes (e.g. criticality or Kolmogorov Complexity) [5,6].

One common approach to measure Type 1 complexity is by measuring entropy, which generally assess the degree of unpredictability or randomness within a signal. These measures can provide valuable information about the underlying dynamical processes of the brain with high entropy signals occurring in states of wakefulness, while low entropy signals can be observed during deep sleep or anesthesia [10].

However, Type 1 complexity as measured via entropy does not capture the full spectrum of complexity, as a purely random process—despite being highly entropic—lacks structure and meaningful organization. High entropy alone does not necessarily indicate maximal complexity in brain function. In contrast, Type 2 complexity addresses this limitation by highlighting that maximal complexity arises in systems that achieve an optimal balance between order and randomness. According to criticality theory, this critical point characterizes the state of maximal computational efficiency and optimal information processing. Such states, often associated with criticality, enable the brain to balance stability and flexibility, a hallmark of higher-order cognitive processes [1-3].

c. The journal style of presenting Methods at the end left this reviewer feeling a lot of information was missing, such as an experimental paradigm and qualitative explanation of algorithms used. At a minimum, the authors could make cross-references to Methods at appropriate places; however, some explanations in the Intro and results would be more reader-friendly.

We have added an overview of our experimental approach and analysis methods at key points in the manuscript. The Introduction now frames our methodological choices and expected findings, while the Results section begins with a clear summary of data collection, preprocessing, and analysis steps. Cross-references to detailed methods sections are included throughout the Results to help readers locate additional methodological information when needed.

By contrasting the effects of caffeine and placebo using inferential statistics and machine learning (ML) separately for non-REM (NREM) and REM sleep, as well as for young and middle-aged groups of adults, we provide the first evidence that caffeine induces a broad increase in EEG brain entropy and a shift towards critical dynamics during sleep. These effects were more widespread in NREM compared to REM sleep, while age-related differences were observed exclusively in REM. To ensure robustness of our results, we provide a comparison of several different metrics of entropy and criticality. Collectively, these findings advance our understanding of how caffeine modulates brain dynamics across different sleep stages and age groups, while offering methodological insights into the assessment of brain signal complexity and criticality under the effect of psychostimulants.

Sleep EEG data were collected from 40 healthy participants during two non-consecutive nights under caffeine and placebo conditions. After preprocessing and artifact removal, we extracted a range of features from the EEG, including power spectral density (PSD), entropy measures, and complexity metrics, to capture caffeine-induced changes in brain activity (see Section 3.5). These features were analyzed both statistically (see Section 3.6) and using supervised ML classifiers (see Section 3.7) to identify differences between conditions. A summary of Cohen's d values for statistical results can be found in Table S3. The detailed experimental design, feature extraction, and analysis pipelines are described in the Methods section. In the following we will go over the results of the spectral power analysis, followed by complexity and criticality-related observations before covering age-related effects.

d. Whereas the extensive use of algorithms has its appeal, I could also see why the remarkably similar effects of the entropy measures could be treated and presented differently, e.g., by stating that 'the measures were correlated and because of being strongly correlated and showing similar effects (Supplementary figure xx), we focus on yy'. If some of the numerous topographic maps were moved to the supplementary material, there would be more figure space for showing the values with the benefit of allowing the reader to evaluate the variability of these measures and the consistency of the effect of caffeine.

We maintain that showing results from multiple entropy metrics strengthens our findings by demonstrating their consistency across different analytical approaches. The complementary nature of spectral and time-domain entropy measures provides unique insights into brain dynamics. While we acknowledge the dense presentation of topographic maps, this comprehensive visualization helps establish the robustness of our findings across different analytical methods. We have improved the statistical analysis of age-related effects in the revised Figure 5 to enhance clarity and precision.

Minor comments:

1. I miss n-numbers on figures for easier readability, e.g., fig. 5.

We have added sample size information for both age groups (young: n=22, middle-aged: n=18) to Figure 5 for clarity. The other figures use the complete dataset (n=40).

2. I think it is not very convincing that there are age, considering the lower statistical power. Besides, just because a channel reaches significance in one group, but not another does not imply that there is a significant effect of age. Thus, I would either use statistics to formally test the effect of age or downplay the robustness of the differences and publish Fig. 5 as a supplementary figure. I certainly would not say that the data support the conclusion in the Discussion, "the brains of the middle-aged individuals were strikingly less perturbed by caffeine than those of the younger subjects," and suggest not to highlight the age effect in the title.

Thanks to the reviewer's valuable input, we have strengthened our statistical analysis of age-related effects. The revised analysis reveals significant age differences specifically in REM sleep for entropy measures and DFA scaling exponent, while no significant age effects were found during NREM sleep. We have updated Figure 5 to clearly display these statistical comparisons between age groups, using independent t-tests to analyze caffeine-placebo differences between young and middle-aged participants. The manuscript now accurately reflects these findings throughout, particularly in the Results and Discussion sections.

We additionally analyzed age-related differences by comparing young (20-27 years) and middle-aged (41-58 years) subgroups using the same paired t-test approach between caffeine versus placebo effects within each age group. To examine age-related differences in caffeine response, we conducted independent t-tests comparing the caffeine-placebo difference between age groups ($(\text{caffeine}_{\text{young}} - \text{placebo}_{\text{young}})$ vs. $(\text{caffeine}_{\text{middle-aged}} - \text{placebo}_{\text{middle-aged}})$). Statistical significance was corrected for multiple comparisons via the FDR Benjamini/Hochberg method and evaluated at $p < 0.05$ and $p < 0.01$.

Previous research has shown that middle-aged adults are more sensitive to caffeine's effects on sleep latency, duration, and efficiency [9], but age-related differences in electrophysiological sleep features remain less understood. In our study, we observed that caffeine had a significantly greater impact on REM sleep EEG features (specifically SpecEn, SampEn and DFA scaling exponent) in younger participants compared to middle-aged adults, while no significant age effects were found during NREM sleep.

Taken together, these findings suggest that caffeine's greater impact on younger adults' REM sleep EEG features arises from an age-dependent interplay between adenosine signaling, receptor density, and caffeine's pharmacological action. Future studies should further explore these interactions, particularly with respect to regional and receptor subtype-specific variations in adenosine activity, to better understand the nuanced effects of caffeine on sleep across the lifespan.

Fig 5. Topographical maps showing age-related differences in brain responses to caffeine versus placebo. The "young" (20-27 years) and "middle-aged" (41-58 years) columns display t-values of paired T-tests between caffeine vs. placebo. To rule out statistical effects of sample size, the remaining columns show the results of independent T-tests (t- and p-values) between age groups $((\text{caffeine}_{\text{young}} - \text{placebo}_{\text{young}}) \text{ vs. } (\text{caffeine}_{\text{middle-aged}} - \text{placebo}_{\text{middle-aged}}))$. Grey dots indicate $p < 0.05$, white dots $p < 0.01$.

3. *The conclusion in the last sentence of the abstract is too vague. Write the novel insight and what implications this has instead of leaving this up to the reader.*

We have revised the abstract's conclusion to directly state our key finding. This clearer statement connects our empirical observations to their theoretical implications for understanding caffeine's effects on neural dynamics.

Interpreting these data in the light of modeling and empirical work on EEG-derived measures of excitation-inhibition balance reveals that caffeine promotes a shift in brain dynamics towards increased neural excitation and closer proximity to a critical regime, particularly during NREM sleep.

4. *I don't understand why the 1st column in Fig. 1 differs from the 1st column of Fig. 2.*

We have corrected the inconsistency between Figures 1 and 2 by updating Figure 1 to match the latest analysis results. The previous minor discrepancy was due to an older version of the analysis script being used for Figure 1. All data now consistently reflect the correct and final analysis outcomes.

5. *If I understand correctly, the authors applied DFA to the "raw" bandpass filtered data as opposed to the amplitude envelope used in the literature referenced in Hardstone et al. 2012. That is very surprising, and the DFA exponents are also usually high. Considering that the DFA was used in such an untraditional fashion, it would seem natural to show the grand average fluctuation functions that the DFA exponents were fit to. The time scales of the authors fit the DFA are also not reported.*

We have clarified our DFA methodology and added extensive detail to the analysis of temporal correlations. Our approach applied DFA to both raw signals and Hilbert envelopes across multiple frequency bands, using standardized epoch lengths and window sizes. The supplementary figure now shows the comparison between envelope-based and raw signal DFA results. While both approaches yield valuable insights, the raw signal analysis provided enhanced sensitivity while maintaining consistency with envelope-based measures in the broadband condition.

We also analyzed the DFA scaling exponents of canonical frequency bands after computing the Hilbert transform. The frequency bands used were the same as in the PSD analysis, and we additionally calculated a broadband DFA in the range of 3 to 32 Hz. While our main focus is on DFA computed from the raw signal, these additional results are provided in Supplementary Figure S4.

Fig S4. DFA on envelope versus raw. t-values comparing statistical findings of detrended fluctuation analysis (DFA) applied to the signal's envelope in contrast to the raw data. We investigate the effect of narrow-band filtering before computing envelopes and compare to broadband DFA. Grey dots represent significance at $p < 0.05$, white dots indicate $p < 0.01$, all corrected for multiple comparisons using maximum statistics.

Since our study's primary goal is comparing complexity and criticality metrics, we maintain our focus on these analyses rather than expanding the DFA results with fluctuation function plots. The existing methodology and results effectively support our main research objectives.

We thank Reviewer 2 for these helpful comments. We feel that the changes we have made to address them have significantly improved our manuscript.

Reviewer #3:

Thoelke et al. reanalyze an EEG sleep dataset (N=40) of participants under prior caffeine consumption vs. placebo. Analyses are performed separately for REM and NREM epochs of 20 seconds duration. They report that caffeine reduces low frequency power in NREM sleep and, alongside, leads to an increase in several complexity measures. The study recovers well known effects of caffeine on sleep electrophysiology, some of which is cited by the authors.

- My main concern is with the novelty of the reported findings. The authors report a reduction of low frequency power in NREM with caffeine, which is well known and a general sign of less slow wave sleep (SWS). It has been shown in several studies that SWS reduces EEG complexity. With reduction of SWS under caffeine it is thus expected that complexity (e.g. measured by entropy, DFA, ...) is relatively increased. Thus, I am hesitant to see the novelty this study provides as all reported effects are essentially only a consequence of the reduced SWS induced by caffeine, which is a well known effect.

We appreciate the reviewer's insights and thoughts about what actually constitutes the novelty of our study. We agree with the reviewer that previous research has established that caffeine induces changes in sleep architecture (esp. reduction in SWS) and that EEG complexity is weakest during SWS. Taken together, these

observations would predict that one of the effects of caffeine would indeed be a reduction of complexity in NREM. We thus understand the reviewer's hesitation and the need to better explain to the readers the significance of our findings.

First of all, although some of our results (e.g. drop in complexity in NREM under caffeine) could have been predicted from previous research, our data are the first scientific evidence which directly measures this effect. Second, by contrast to most of the previous research (which has often focused on changes in spectral power), we frame our investigation in the larger context of brain criticality and complexity. Third, we used a machine learning approach that jointly explores a wide array of features and determines which ones capture the most prominent changes induced by caffeine (multi-feature classification and feature importance assessment). This specific analysis has for instance uncovered that the complexity metrics contribute more to the discrimination between the caffeine and placebo conditions in the sleep EEG, than the power spectral features. Related to this, our criticality-related metrics demonstrate for the first time that caffeine shifts the NREM brain dynamics closer to a critical regime. In addition to these analyses, our study specifically addresses the potential effects of age by comparing the observations across young and middle-aged individuals, revealing potentially interesting significant differences in REM sleep.

Because we believe that the reviewer's point about the need to clarify these novelties is important, we have substantially revised the narrative and presentation of the rationale. More specifically, we have expanded our introduction and discussion sections to better highlight previously known effects, and how they led to our working hypotheses and our findings. In addition, the limitations linked to caffeine-induced changes in sleep structure are now more explicitly addressed.

[...] caffeine is known to alter sleep architecture by reducing slow-wave sleep (SWS) and increasing lighter sleep stages such as N1 and N2 [8,9]. Complexity measures, including entropy, have been shown to reliably reflect these changes, with entropy being highest in wakefulness, followed by REM sleep, and progressively decreasing across N1, N2, and SWS [10,11]. We expect that caffeine-induced alterations in sleep architecture, specifically the reduction of SWS and the increase in lighter sleep stages (N1 and N2), will result in a measurable increase in EEG complexity during NREM sleep.

Finally, caffeine is known to alter the ratio of NREM stages (N1, N2, SWS) to each other, making it difficult to disentangle the effect of caffeine on individual NREM stages from the shift in sleep architecture from deeper to lighter sleep. Many of the caffeine-related significant changes we found in NREM also occur during the shift from deeper to lighter sleep. Thus, further work should look into this.

- There is also some work on using ML to classify epochs under caffeine from controls using these features (power, complexity etc.)

We have conducted a thorough literature review and found no machine learning studies classifying sleep EEG epochs between caffeine and control conditions. We

welcome any specific references the reviewer could share to help us address these missing references. We'd be happy to add new references and compare our observations to previous ML work comparing the effect of caffeine and placebo on sleep EEG.

- Minor: Scoring sleep stages: the authors use S3, S4 stages -> now according to new classification just S3 (or N3) should be used. Since the authors combine all NREM stages into one class, this does not matter too much.

We acknowledge that our sleep stage scoring followed historical conventions using S1-S4 stages, as this data was collected before AASM guidelines became widely adopted. The reviewer is right that this methodological difference does not impact our findings because we analyzed NREM sleep as a single stage. To make sure this is clear to all readers, we have updated the methods section to more clearly explain this distinction.

The data was divided into 20s windows and visually scored by an expert into five sleep stages: S1, S2, S3, S4, and REM according to the Rechtschaffen and Kales manual [48], modified to allow scoring based on 20s epochs [9]. While scoring sleep into S1, S2, S3, and S4 is no longer standard practice and has since been replaced by the AASM guidelines which consolidate S3 and S4 into a single N3 stage, this approach reflects the common scoring practice at the time of data collection. Importantly, this distinction does not impact the study's conclusions, as we combined S1, S2, S3, and S4 into a single non-REM (NREM) stage for analysis.

- Minor: Fig.2 b, please add units to axes. Is the y-axis in log-scale?

We appreciate the careful review of our figures. Your observation is correct - Figure 2b presents data in log-log scale. We deliberately omitted axis labels and tick marks since this figure serves purely as a visual illustration of the aperiodic slope effect using data from a single subject and channel. Adding quantitative markers could mislead readers into drawing conclusions from this limited example, which is intended only to aid conceptual understanding. As other readers might have the same question, we have clarified the nature of this figure panel in the associated caption.

Fig 2. [...] Illustrative example of caffeine-induced shifts in aperiodic slope (solid lines), showing full power spectra (dashed lines) from a single subject at electrode Fz (red star) in a log-log plot. Subject and channel were chosen to be representative for the effect found across subjects. This panel does not contain results of statistical tests but rather serves as a clarification to the reader about the effect of the aperiodic slope on power spectral density across frequency bands.

We would like to sincerely thank Reviewer 3 for their insightful comments which have definitely helped us modify the positioning of the study and better highlight its added value and contribution to the field.

References

1. Toker, D. et al. Consciousness is supported by near-critical slow cortical electro-dynamics. *Proceedings of the National Academy of Sciences* 119, e2024455119 (2022). URL <https://www.pnas.org/doi/10.1073/pnas.2024455119>. Publisher: Proceedings of the National Academy of Sciences.
2. Kinouchi, O. & Copelli, M. Optimal dynamical range of excitable networks at criticality. *Nature Physics* 2, 348–351 (2006). URL <https://www.nature.com/articles/nphys289>. Publisher: Nature Publishing Group.
3. O’Byrne, J. & Jerbi, K. How critical is brain criticality? *Trends in Neurosciences* (2022). URL <https://www.sciencedirect.com/science/article/pii/S0166223622001643>.
4. Linkenkaer-Hansen, K., Nikouline, V. V., Palva, J. M. & Ilmoniemi, R. J. Long-Range Temporal Correlations and Scaling Behavior in Human Brain Oscillations. *Journal of Neuroscience* 21, 1370–1377 (2001). URL <https://www.jneurosci.org/content/21/4/1370>. Publisher: Society for Neuroscience Section: ARTICLE.
5. Gell-Mann, M. & Lloyd, S. Information measures, effective complexity, and total information. *Complexity* 2, 44–52 (1996). URL <https://onlinelibrary.wiley.com/doi/abs/10.1002/%28SICI%291099-0526%28199609/10%292%3A1%3C44%3A%3AAID-CPLX10%3E3.0.CO%3B2-X>. eprint: <https://onlinelibrary.wiley.com/doi/pdf/10.1002/%28SICI%291099-0526%28199609/10%292%3A1%3C44%3A%3AAID-CPLX10%3E3.0.CO%3B2-X>.
6. Zanin, M. & Olivares, F. Ordinal patterns-based methodologies for distinguishing chaos from noise in discrete time series. *Communications Physics* 4, 1–14 (2021). URL <https://www.nature.com/articles/s42005-021-00696-z>. Publisher: Nature Publishing Group.
7. Chang, D. et al. Caffeine Caused a Widespread Increase of Resting Brain Entropy. *Scientific Reports* 8, 1–7 (2018). URL <https://www.nature.com/articles/s41598-018-21008-6>. Number: 1 Publisher: Nature Publishing Group.
8. Drapeau, C. et al. Challenging sleep in aging: the effects of 200 mg of caffeine during the evening in young and middle-aged moderate caffeine consumers. *Journal of Sleep Research* 15, 133–141 (2006). URL <https://onlinelibrary.wiley.com/doi/abs/10.1111/j.1365-2869.2006.00518.x>. eprint: <https://onlinelibrary.wiley.com/doi/pdf/10.1111/j.1365-2869.2006.00518.x>.
9. Robillard, R., Bouchard, M., Cartier, A., Nicolau, L. & Carrier, J. Sleep is more sensitive to high doses of caffeine in the middle years of life. *Journal of Psychopharmacology* 29, 688–697 (2015). URL <https://doi.org/10.1177/0269881115575535>. Publisher: SAGE Publications Ltd STM.

10. Bruhn, J., Röpcke, H., Rehberg, B., Bouillon, T. & Hoefft, A. Electroencephalogram approximate entropy correctly classifies the occurrence of burst suppression pattern as increasing anesthetic drug effect. *Anesthesiology* 93, 981–985 (2000).
11. Bruce, E. N., Bruce, M. C. & Vennelaganti, S. Sample Entropy Tracks Changes in EEG Power Spectrum With Sleep State and Aging. *Journal of clinical neurophysiology : official publication of the American Electroencephalographic Society* 26, 257–266 (2009). URL <https://www.ncbi.nlm.nih.gov/pmc/articles/PMC2736605/>.
12. Landolt, H.-P. & Borbély, A. A. Age-dependent changes in sleep EEG topography. *Clinical Neurophysiology* 112, 369–377 (2001). URL <https://www.sciencedirect.com/science/article/pii/S1388245700005423>.
13. Voytek, B. et al. Age-Related Changes in 1/f Neural Electrophysiological Noise. *Journal of Neuroscience* 35, 13257–13265 (2015). URL <https://www.jneurosci.org/content/35/38/13257>. Publisher: Society for Neuroscience Section: Articles.
14. Thuwal, K., Banerjee, A. & Roy, D. Aperiodic and Periodic Components of Ongoing Oscillatory Brain Dynamics Link Distinct Functional Aspects of Cognition across Adult Lifespan. *eNeuro*, ENEURO.0224–21.2021 (2021). URL <https://www.ncbi.nlm.nih.gov/pmc/articles/PMC8547598/>.
15. Medel, V., Irani, M., Ossandon, T. & Boncompte, G. Complexity and 1/f slope jointly reflect cortical states across different E/I balances (2020). URL <https://www.biorxiv.org/content/10.1101/2020.09.15.298497v2>. Pages: 2020.09.15.298497 Section: New Results.
16. Lombardi, F., Herrmann, H. J. & de Arcangelis, L. Balance of excitation and inhibition determines 1/f power spectrum in neuronal networks. *Chaos: An Interdisciplinary Journal of Nonlinear Science* 27, 047402 (2017). URL <https://aip.scitation.org/doi/abs/10.1063/1.4979043>. Publisher: American Institute of Physics.
17. Pincus, S. M. Greater signal regularity may indicate increased system isolation. *Mathematical Biosciences* 122, 161–181 (1994). URL <https://linkinghub.elsevier.com/retrieve/pii/0025556494900566>.
18. Massimini, M. et al. Breakdown of Cortical Effective Connectivity During Sleep. *Science* 309, 2228–2232 (2005). URL <https://www.science.org/doi/10.1126/science.1117256>. Publisher: American Association for the Advancement of Science.
19. Massimini, M. et al. Triggering sleep slow waves by transcranial magnetic stimulation. *Proceedings of the National Academy of Sciences* 104, 8496–8501 (2007). URL <https://www.pnas.org/doi/10.1073/pnas.0702495104>. Publisher: Proceedings of the National Academy of Sciences.

20. Lee, G., Fattinger, S., Mouthon, A.-L., Noirhomme, Q. & Huber, R. Electroencephalogram approximate entropy influenced by both age and sleep. *Frontiers in Neuroinformatics* 7 (2013). URL <https://www.frontiersin.org/article/10.3389/fninf.2013.00033>.
21. Keshmiri, S. Entropy and the Brain: An Overview. *Entropy* 22, 917 (2020). URL <https://www.mdpi.com/1099-4300/22/9/917>. Number: 9 Publisher: Multidisciplinary Digital Publishing Institute.
22. Zimmern, V. Why Brain Criticality Is Clinically Relevant: A Scoping Review. *Frontiers in Neural Circuits* 14 (2020). URL <https://www.frontiersin.org/journals/neural-circuits/articles/10.3389/fncir.2020.00054/full>. Publisher: Frontiers.
23. Jamasebi, R., Redline, S., Patel, S. R. & Loparo, K. A. Entropy-based Measures of EEG Arousals as Biomarkers for Sleep Dynamics: Applications to Hypertension. *Sleep* 31, 935–943 (2008). URL <https://doi.org/10.5665/sleep/31.7.935>.
24. Azami, H. et al. EEG entropy in REM sleep as a physiologic biomarker in early clinical stages of Alzheimer's disease. *Journal of Alzheimer's disease : JAD* 91, 1557–1572 (2023). URL <https://www.ncbi.nlm.nih.gov/pmc/articles/PMC10039707/>.
25. Bjorness, T. E. & Greene, R. W. Adenosine and Sleep. *Current Neuropharmacology* 7, 238–245 (2009).
26. Meyer, P. T. et al. Effect of aging on cerebral A1 adenosine receptors: A [18F]CPFPX PET study in humans. *Neurobiology of Aging* 28, 1914–1924 (2007). URL <https://www.sciencedirect.com/science/article/pii/S0197458006003009>.
27. Gao, R., Peterson, E. J. & Voytek, B. Inferring synaptic excitation/inhibition balance from field potentials. *NeuroImage* 158, 70–78 (2017). URL <https://www.sciencedirect.com/science/article/pii/S1053811917305621>.
28. Martínez-Cañada, P. & Panzeri, S. Mahmud, M., Kaiser, M. S., Vassanelli, S., Dai, Q. & Zhong, N. (eds) Spectral Properties of Local Field Potentials and Electroencephalograms as Indices for Changes in Neural Circuit Parameters. (eds Mahmud, M., Kaiser, M. S., Vassanelli, S., Dai, Q. & Zhong, N.) *Brain Informatics, Lecture Notes in Computer Science*, 115–123 (Springer International Publishing, Cham, 2021).
29. Trakoshis, S. et al. Intrinsic excitation-inhibition imbalance affects medial prefrontal cortex differently in autistic men versus women. *eLife* 9, e55684 (2020). URL <https://doi.org/10.7554/eLife.55684>. Publisher: eLife Sciences Publications, Ltd.
30. Stadnitski, T. Measuring Fractality. *Frontiers in Physiology* 3 (2012). URL <https://www.frontiersin.org/articles/10.3389/fphys.2012.00127>.
31. Porkka-Heiskanen, T. & Kalinchuk, A. V. Adenosine, energy metabolism and sleep homeostasis. *Sleep Medicine Reviews* 15, 123–135 (2011). URL <https://www.sciencedirect.com/science/article/pii/S1087079210000663>.

32. Ahmad, J. et al. From mechanisms to markers: novel noninvasive EEG proxy markers of the neural excitation and inhibition system in humans. *Translational Psychiatry* 12, 1–12 (2022). URL <https://www.nature.com/articles/s41398-022-02218-z>. Number: 1 Publisher: Nature Publishing Group.
33. Porjesz, B. et al. Linkage disequilibrium between the beta frequency of the human EEG and a GABAA receptor gene locus. *Proceedings of the National Academy of Sciences* 99, 3729–3733 (2002). URL <https://www.pnas.org/doi/10.1073/pnas.052716399>. Publisher: Proceedings of the National Academy of Sciences.
34. Jensen, O. et al. On the human sensorimotor-cortex beta rhythm: sources and modeling. *NeuroImage* 26, 347–355 (2005).
35. Ribeiro, J. A. & Sebastião, A. M. Caffeine and adenosine. *Journal of Alzheimer's disease: JAD* 20 Suppl 1, S3–15 (2010).
36. Beggs, J. M. & Plenz, D. Neuronal Avalanches in Neocortical Circuits. *Journal of Neuro-science* 23, 11167–11177 (2003). URL <https://www.jneurosci.org/content/23/35/11167>. Publisher: Society for Neuroscience Section: Behavioral/Systems/Cognitive.
37. Shew, W. L., Yang, H., Petermann, T., Roy, R. & Plenz, D. Neuronal Avalanches Imply Maximum Dynamic Range in Cortical Networks at Criticality. *Journal of Neuroscience* 29, 15595–15600 (2009). URL <https://www.jneurosci.org/content/29/49/15595>. Publisher: Society for Neuroscience Section: Brief Communications.
38. Botcharova, M., Farmer, S. F. & Berthouze, L. Markers of criticality in phase synchronization. *Frontiers in Systems Neuroscience* 8 (2014). URL <https://www.frontiersin.org/journals/systems-neuroscience/articles/10.3389/fnsys.2014.00176/full>. Publisher: Frontiers.
39. Peng, C.-K. et al. Mosaic organization of DNA nucleotides. *Physical Review E* 49, 1685–1689 (1994). URL <https://link.aps.org/doi/10.1103/PhysRevE.49.1685>. Publisher: American Physical Society.
40. Hardstone, R. et al. Detrended Fluctuation Analysis: A Scale-Free View on Neuronal Oscillations. *Frontiers in Physiology* 3 (2012). URL <https://www.frontiersin.org/articles/10.3389/fphys.2012.00450>.
41. Vallat, R. [raphaelvallat/antropy](https://github.com/raphaelvallat/antropy) (2024). URL <https://github.com/raphaelvallat/antropy>. Original-date: 2021-03-17T00:51:14Z.
42. Donoghue, T. et al. Parameterizing neural power spectra into periodic and aperiodic components. *Nature Neuroscience* 23, 1655–1665 (2020). URL <https://www.nature.com/articles/s41593-020-00744-x>. Number: 12 Publisher: Nature Publishing Group.

43. Nichols, T. E. & Holmes, A. P. Nonparametric permutation tests for functional neuroimaging: a primer with examples. *Human Brain Mapping* 15, 1–25 (2002).
44. Pantazis, D., Nichols, T. E., Baillet, S. & Leahy, R. M. A comparison of random field theory and permutation methods for the statistical analysis of MEG data. *NeuroImage* 25, 383–394 (2005).
45. Ojala, M. & Garriga, G. C. Permutation Tests for Studying Classifier Performance. *The Journal of Machine Learning Research* 11, 1833–1863 (2010).
46. Good, P. *Permutation Tests: A Practical Guide to Resampling Methods for Testing Hypotheses* (Springer Science & Business Media, 2013). Google-Books-ID: pK3hBwAAQBAJ.
47. Thölke, P. et al. Class imbalance should not throw you off balance: Choosing the right classifiers and performance metrics for brain decoding with imbalanced data. *NeuroImage* 277, 120253 (2023). URL <https://www.sciencedirect.com/science/article/pii/S1053811923004044>.
48. Allan Rechtschaffen, A. K. *A Manual of Standardized Terminology, Techniques and Scoring System for Sleep Stages of Human Subjects* (U. S. National Institute of Neurological Diseases and Blindness, Neurological Information Network, 1968). URL <http://archive.org/details/RKManual>.

Ms. No.: COMMSBIO-24-3851B

Title: Caffeine induces age-dependent increases in brain complexity and criticality during sleep

In the following, we provide point-by-point responses to all the comments and questions raised by Reviewers #1, #2 and #3, from a previous submission of this study to Nature Communications Biology (COMMSBIO-24-3851B). For the sake of clarity, the Reviewers' comments are in italic text, our responses are in bold font, and additions/changes made to the manuscript are presented here (as well as in the revised manuscript) in blue.

Reviewer #1:

The updated manuscript is significantly improved, particularly with regard to methodology. Some minor improvements could still be made regarding the points below:

We appreciate the reviewer's positive feedback on the revised manuscript.

1) When it comes to the age related effects, some interpretation of the differences between younger and middle aged participants is needed. For instance, how might baseline differences in sleep architecture or receptor density influence the observed effects? Could age-related cognitive or lifestyle factors play a role?

We agree that further clarifications would be useful here. We hypothesize that a high baseline adenosine A₁ receptor density likely boosts the strength of caffeine's effects on the brain. This is one relevant section from the present manuscript:

In younger adults, higher A₁ receptor density may amplify the effects of caffeine, which acts by blocking adenosine binding to these receptors. During REM sleep, this interaction could result in a compound effect of reduced adenosine activity (characteristic of REM) and higher receptor availability, enabling caffeine to exert a more pronounced influence on brain dynamics. By contrast, the diminished A₁ receptor density in middle-aged adults likely limits caffeine's impact, as fewer receptors are available for adenosine binding and subsequent blockade by caffeine.

We have further added the following discussion about the effects of baseline sleep architecture and aging/lifestyle factors:

Beyond receptor mechanisms, several factors may explain age-related differences in caffeine sensitivity during REM sleep. Age-related changes in caffeine metabolism [1] likely play a role, as decreased hepatic clearance in older adults can alter caffeine's concentration

during different sleep phases, potentially explaining the attenuated effects we observed in middle-aged participants. Baseline sleep architecture differences may also contribute—middle-aged adults typically experience reduced REM sleep quantity and quality [2]—potentially creating a ceiling effect where already compromised REM sleep shows less disruption from caffeine. Additionally, age-specific lifestyle factors such as work stress, family responsibilities, and exercise habits may modulate baseline sleep characteristics and caffeine responses through indirect pathways involving stress hormones and inflammatory markers [3].

Taken together, these findings suggest that caffeine's greater impact on younger adults' REM sleep EEG features arises from an age-dependent interplay between adenosine signaling, receptor density, caffeine metabolism, and caffeine's pharmacological action. Age-related differences in hepatic clearance and baseline sleep architecture likely contribute to the attenuated response in middle-aged adults. Future studies should further explore these interactions, particularly with respect to regional and receptor subtype-specific variations in adenosine activity, caffeine pharmacokinetics, and lifestyle factors, to better understand the nuanced effects of caffeine on sleep across the lifespan.

2) There is a strong focus on healthy individuals - but how might the findings be generalized to larger populations with sleep disorders or neurodegenerative conditions?

We expanded the “Limitations” section with the following paragraph regarding clinical populations:

Finally, our study focused on healthy individuals, which may limit the generalizability of our findings. Altered baseline brain dynamics and sleep architecture associated with sleep disorders and neurodegenerative diseases may interact with caffeine's effects on the brain in complex ways. While regular daytime caffeine consumption has been associated with neuroprotective qualities, particularly in the context of Parkinson's disease [4-7], acute caffeine intake close to bedtime is known to disrupt sleep, potentially affecting critical brain processes that occur during sleep. For example, caffeine could hypothetically exacerbate sleep fragmentation already present in Alzheimer's [8] and Parkinson's disease [9], though our study does not provide empirical evidence for this specific interaction. Given the complex interplay between caffeine's beneficial neuroprotective effects, its disruptive influence on sleep, and the sleep physiology changes observed with aging, future research should examine how caffeine-induced alterations in sleep brain dynamics manifest across different clinical populations. Such studies could help inform tailored caffeine consumption recommendations for individuals with neurological disorders.

3) the potential caffeine induced shifts in nrem architecture is given as a limitation - how might such shifts influence complexity and criticality?

In the “Aim and hypotheses” section of the Introduction in the manuscript, we suggest that the altered NREM sleep architecture may lead to an increase in EEG complexity due to a shift from deeper toward lighter sleep:

Furthermore, caffeine is known to alter sleep architecture by reducing slow-wave sleep (SWS) and increasing lighter sleep stages such as N1 and N2 [10,11]. Complexity

measures, including entropy, have been shown to reliably reflect these changes, with entropy being highest in wakefulness, followed by REM sleep, and progressively decreasing across N1, N2, and SWS [12,13]. We expect that caffeine-induced alterations in sleep architecture, specifically the reduction of SWS and the increase in lighter sleep stages (N1 and N2), will result in a measurable increase in EEG complexity during NREM sleep.

We further modified our paragraph on this limitation in the discussion section:

Fourth, caffeine is known to alter the ratio of NREM stages (N1, N2, SWS) to each other, making it difficult to disentangle the effect of caffeine on individual NREM stages from the caffeine-induced shift in sleep architecture (less SWS, more N1/N2). Many of the caffeine-related significant changes we found in NREM are characteristic for such a shift. Specifically, hallmark features of SWS, such as low entropy/complexity, will contribute less to the aggregated measures investigated in this study.

We sincerely thank Reviewer 1 for their thoughtful questions and suggested refinements.

Reviewer #2:

The authors have addressed my previous comments and suggestions well.

We sincerely thank Reviewer 2 for their helpful and constructive feedback in the previous round and for recognizing our efforts in addressing the comments. We truly appreciate the time and care devoted to evaluating our work.

References

1. Tanaka. In vivo age-related changes in hepatic drug-oxidizing capacity in humans. *Journal of Clinical Pharmacy and Therapeutics* 23, 247-255 (1998).
2. Ohayon, M. M., Carskadon, M. A., Guilleminault, C. & Vitiello, M. V. Meta-Analysis of Quantitative Sleep Parameters From Childhood to Old Age in Healthy Individuals: Developing Normative Sleep Values Across the Human Lifespan. *Sleep* 27, 1255-1273 (2004).
3. Meerlo, P., Sgoifo, A. & Suchecki, D. Restricted and disrupted sleep: Effects on autonomic function, neuroendocrine stress systems and stress responsivity. *Sleep Medicine Reviews* 12, 197-210 (2008).
4. Frigerio, R. et al. Comparison of Risk Factor Profiles in Incidental Lewy Body Disease and Parkinson Disease. *Archives of Neurology* 66, 1114-1119 (2009).
5. Ren, X. & Chen, J.-F. Caffeine and Parkinson's Disease: Multiple Benefits and Emerging Mechanisms. *Frontiers in Neuroscience* 14 (2020).
6. Palacios, N. et al. Caffeine and risk of Parkinson's disease in a large cohort of men and women. *Movement Disorders* 27, 1276-1282 (2012).
7. Ross, G. W. et al. Association of Coffee and Caffeine Intake With the Risk of Parkinson Disease. *JAMA* 283, 2674-2679 (2000).
8. Mander, B. A., Winer, J. R., Jagust, W. J. & Walker, M. P. Sleep: A Novel Mechanistic Pathway, Biomarker, and Treatment Target in the Pathology of Alzheimer's Disease? *Trends in Neurosciences* 39, 552-566 (2016).
9. Comella, C. L. Sleep disorders in Parkinson's disease: An overview. *Movement Disorders* 22, S367-S373 (2007).
10. Drapeau, C. et al. Challenging sleep in aging: the effects of 200 mg of caffeine during the evening in young and middle-aged moderate caffeine consumers. *Journal of Sleep Research* 15, 133-141 (2006).
11. Robillard, R., Bouchard, M., Cartier, A., Nicolau, L. & Carrier, J. Sleep is more sensitive to high doses of caffeine in the middle years of life. *Journal of Psychopharmacology* 29, 688-697 (2015).
12. Bruhn, J., Röpcke, H., Rehberg, B., Bouillon, T. & Hoefft, A. Electroencephalogram approximate entropy correctly classifies the occurrence of burst suppression pattern as increasing anesthetic drug effect. *Anesthesiology* 93, 981-985 (2000).
13. Bruce, E. N., Bruce, M. C. & Vennelaganti, S. Sample Entropy Tracks Changes in EEG Power Spectrum With Sleep State and Aging. *Journal of clinical*

neurophysiology: official publication of the American Electroencephalographic Society 26, 257-266 (2009).